# Nearly Dimension-Independent Convergence of Mean-Field Black-Box Variational Inference

**Kyurae Kim**
University of Pennsylvania
kyrkim@seas.upenn.edu

**Yi-An Ma**
University of California San Diego
yianma@ucsd.edu

**Trevor Campbell**
University of British Columbia
trevor@stat.ubc.ca

**Jacob R. Gardner**
University of Pennsylvania
jacobrg@seas.upenn.edu

## Abstract

We prove that, given a mean-field location-scale variational family, black-box variational inference (BBVI) with the reparametrization gradient converges at a rate that is nearly independent of any explicit dimension dependence. Specifically, for a $d$-dimensional strongly log-concave and log-smooth target, the number of iterations for BBVI with a sub-Gaussian family to obtain a solution $\epsilon$-close to the global optimum has an explicit dimension dependence no larger than $\mathrm{O}(\log d)$. This is a significant improvement over the $\mathrm{O}(d)$ dependence of full-rank location-scale families. For heavy-tailed families, we prove a weaker $\mathrm{O}(d^{2/k})$ dependence, where $k$ is the number of finite moments of the family. Additionally, if the Hessian of the target log-density is constant, the complexity is free of any explicit dimension dependence. We also prove that our bound on the gradient variance, which is key to our result, cannot be improved using only spectral bounds on the Hessian of the target log-density.

## 1 Introduction

Variational inference (VI; Blei et al., 2017; Hinton and van Camp, 1993; Jordan et al., 1999; Peterson and Hartman, 1989) is an effective method for approximating intractable high-dimensional distributions and models with tall datasets. Among various VI algorithms, black-box VI (BBVI; Kucukelbir et al., 2017; Ranganath et al., 2014; Titsias and Lázaro-Gredilla, 2014; Wingate and Weber, 2013), which minimizes the exclusive KL divergence (Kullback and Leibler, 1951) via stochastic gradient descent (SGD; Bottou et al., 2018; Robbins and Monro, 1951) in the space of parameters, is widely used due to its flexibility to apply to a wide range of variational families with only minor modifications (Bingham et al., 2019; Carpenter et al., 2017; Fjelde et al., 2025; Ge et al., 2018; Patil et al., 2010). Specifically, location-scale variational families—in which a base distribution is mutated by an affine transformation—remain a popular choice, encompassing those with diagonal scale matrices (the "mean-field" approximation; Hinton and van Camp, 1993; Peterson and Hartman, 1989), as well as scale matrices with low rank (Ong et al., 2018; Rezende et al., 2014; Tomczak et al., 2020) and full-rank (Kucukelbir et al., 2017; Titsias and Lázaro-Gredilla, 2014) factors.

The choice of the variational family is generally known to affect the convergence speed of BBVI, where families that are more "expressive," those that contain more complex distributions, result in slower convergence. For example, in location-scale families, it has been empirically observed that mean-field families often provide faster convergence to an accurate posterior approximation than full-rank families (Agrawal et al., 2020; Giordano et al., 2018, 2024; Ko et al., 2024; Zhang et al., 2022). This is because full-rank families often require running SGD with a smaller step size and

for longer; even given a large computation budget, BBVI on a full-rank family may not converge adequately (Ko et al., 2024). Therefore, choosing the expressiveness of the family corresponds to trading statistical accuracy for computational efficiency (Bhatia et al., 2022). In order to control this trade-off for our benefit, a clear theoretical understanding of the relationship between convergence speed and expressiveness is needed.

Formally, consider the setting of approximating a $\mu$-strongly log-concave and $L$-log-smooth target, where $\kappa \triangleq L/\mu$ is the condition number. For BBVI with the reparametrization gradient (Kingma and Welling, 2014; Rezende et al., 2014; Titsias and Lázaro-Gredilla, 2014) on a full-rank location-scale family, an $\epsilon$-close solution to the global optimum in squared distance in parameter space can be obtained after at least $O(d\kappa^2\epsilon^{-1})$ iterations (Domke, 2019; Kim et al., 2023a). For mean-field location-scale families, on the other hand, the iteration complexity improves to $O(\sqrt{d}\,\kappa^2\epsilon^{-1})$ (Kim et al., 2023a). While this is clearly better than the $O(d)$ explicit dimension dependence of full-rank families, it has been conjectured that a better dependence is more likely (Kim et al., 2023a).

In this work, we positively resolve this conjecture by obtaining stronger convergence guarantees for BBVI on mean-field location-scale families (Section 3). In particular, under the conditions stated above, we prove that BBVI with a mean-field location-scale family with sub-Gaussian tails can obtain an $\epsilon$-accurate solution in squared distance after $O((\log d)\kappa^2\epsilon^{-1})$ iterations. Heavier-tailed families achieve a weaker $O(d^{2/k}\kappa^2\epsilon^{-1})$ iteration complexity guarantee, where $k$ is the number of finite moments of the variational family. For the Student-$t$ variational family with a high-enough degrees of freedom $\nu$, this corresponds to a $O(d^{2/(\nu-2)})$ explicit dimension dependence. In addition, if the Hessian of the target log-density is constant, any mean-field location-scale family attains a $O(\kappa^2\epsilon^{-1})$ iteration complexity without any explicit dependence on $d$.

The key element of the proof is a careful probabilistic analysis of the variance of the reparametrization gradient (Section 4): In general, the reparametrization gradient of the scale parameters contains heavy-tailed components that grow not-so-slowly in $d$. However, for mean-field families, only a *single* random coordinate turns out to be heavy-tailed. Through a probabilistic decomposition, the influence of this heavy-tailed component can be averaged out over all $d$ coordinates. Then the lighter-tailed components of the gradient dominate as $d$ increases, resulting in a benign dimension dependence (Lemma 4.1). We also provide a lower bound (Proposition 4.2) showing that our analysis cannot be improved when using only spectral bounds on the Hessian of the target log-density.

## 2 Preliminaries

**Notation**  We denote random variables in sans serif (*e.g.*, $\mathsf{u}$, $\mathsf{U}$). $\mathbb{S}^d_{\succ 0} \subset \mathbb{R}^{d\times d}$ denotes the set of $d \times d$ positive definite (PD) matrices, $\mathbb{D}^d \subset \mathbb{R}^{d\times d}$ denotes the set of diagonal matrices, and $\mathbb{D}^d_{\succ 0} \subset \mathbb{D}^d \cap \mathbb{S}^{d\times d}_{\succ 0}$ is its positive definite subset. $\langle \cdot, \cdot \rangle$ and $\|\cdot\|_2$ denote the Euclidean inner product and norm. For a matrix $A \in \mathbb{R}^{d\times d}$, $\|A\|_{\mathrm{F}} = \sqrt{\mathrm{tr}(A^\top A)}$ is the Frobenius norm, $\|A\|_2 = \sigma_{\max}(A)$ is the $\ell_2$ operator norm, where $\sigma_{\max}(\cdot)$ and $\sigma_{\min}(\cdot)$ are the largest and smallest singular values.

### 2.1 Problem Setup

Our problem of interest is an optimization problem over some space $\Lambda \subseteq \mathbb{R}^p$ of the form of

$$\operatorname*{minimize}_{\lambda \in \Lambda} \left\{ F(\lambda) \triangleq f(\lambda) + h(\lambda) \right\}, \quad \text{where} \quad f(\lambda) \triangleq \mathbb{E}_{\mathsf{z}\sim q_\lambda} \ell(\mathsf{z}), \tag{1}$$

$\ell : \mathbb{R}^d \to \mathbb{R}$ is a measurable function we refer to as the "target function", $h : \Lambda \to \mathbb{R}$ is a potentially non-smooth convex regularizer, and the expectation $\mathbb{E}_{\mathsf{z}\sim q_\lambda} \ell(\mathsf{z})$ is assumed to be intractable.

BBVI is a special case of Eq. (1) where $\ell = -\log\pi$ is the negative (unnormalized) log-density of some distribution $\pi$ with respect to the Lebesgue measure and $h(\lambda) = -\mathbb{H}[q_\lambda]$ is the negative differential entropy of $q_\lambda$. Then $F$ is the exclusive Kullback-Leibler divergence $\mathrm{D}_{\mathrm{KL}}$ (Kullback and Leibler, 1951) up to an additive constant (Jordan et al., 1999), where Eq. (1) reduces to

$$\operatorname*{minimize}_{\lambda \in \Lambda} \left\{ \mathrm{D}_{\mathrm{KL}}(q_\lambda, \pi) \propto -\mathbb{E}_{\mathsf{z}\sim q_\lambda} \log\pi(\mathsf{z}) - \mathbb{H}(q_\lambda) \right\}, \tag{2}$$

We assume $\pi$ is supported on $\mathbb{R}^d$, which, unless discrete-valued variables are involved, is often valid after appropriate support transformations (Kim et al., 2023a, §2.2). Such a setup for BBVI has been proposed by Kucukelbir et al. (2017), and now encompasses most practical use of BBVI with the

reparametrization gradient as implemented in Stan (Carpenter et al., 2017), PyMC (Patil et al., 2010), Pyro (Bingham et al., 2019), and Turing (Fjelde et al., 2025; Ge et al., 2018).

For the purpose of a quantitative theoretical analysis, we will consider the following properties:

**Definition** (Smoothness)**.** *For some $\phi : \mathbb{R}^d \to \mathbb{R}$, we say $\phi$ is L-(Lipschitz )smooth if there exists some $L \in (0, +\infty)$ such that, for all $z, z' \in \mathbb{R}^d$,*

$$\|\nabla\phi(z) - \nabla\phi(z')\|_2 \leq L\|z - z'\|_2 \ .$$

**Definition** (Strong Convexity)**.** *For some $\phi : \mathbb{R}^d \to \mathbb{R}$, we say $\phi$ is $\mu$-strongly convex if there exists some constant $\mu \in (0, L]$ such that, for all $z, z' \in \mathbb{R}^d$,*

$$\langle\nabla\phi(z), z - z'\rangle \geq \phi(z) - \phi(z') + \frac{\mu}{2}\|z - z'\|_2^2 \ .$$

In the context of BBVI, assuming that $\ell = -\log\pi$ is both $\mu$-strongly convex and $L$-smooth is equivalent to assuming $\pi$ is $\mu$-strongly log-concave and $L$-log-Lipschitz smooth, respectively, which is common in the analysis of MCMC (Chewi, 2024) and VI (Arnese and Lacker, 2024; Diao et al., 2023; Domke et al., 2023; Kim et al., 2023a; Lambert et al., 2022; Lavenant and Zanella, 2024).

## 2.2 Variational Family

We consider the location-scale family (Casella and Berger, 2001, §3.5):

**Definition 2.1** (Location-Scale Variational Family)**.** *A family of distributions $\mathcal{Q}$ is referred to as a location-scale variational family if there exists some univariate distribution $\varphi$ dominated by the Lebesgue measure such that each member of $\mathcal{Q}$ indexed by $\lambda = (m, C) \in \mathbb{R}^d \times \mathcal{C}$, where $\mathcal{C} \subset \mathbb{R}^{d \times d}$ and $q_\lambda \in \mathcal{Q}$, satisfies*

$$z \sim q_\lambda \qquad \Leftrightarrow \qquad z \overset{\mathrm{d}}{=} \mathcal{T}_\lambda(u) \ ,$$

*where*

$$\mathcal{T}_\lambda(u) \triangleq Cu + m, \qquad u \triangleq (u_1, \ldots, u_d), \qquad u_i \overset{\text{i.i.d.}}{\sim} \varphi \ ,$$

*and $\overset{\mathrm{d}}{=}$ is equivalence in distribution. Then $\mathcal{T}_\lambda$ is referred to as the "reparametrization function," while $m$ and $C$ are referred to as the location and scale parameters, respectively.*

In addition, we impose mild regularity assumptions on the moments of the base distribution:

**Assumption 2.2.** $\varphi$ *satisfies the following: (i) It is standardized such that $\mathbb{E}u_i = 0$ and $\mathbb{E}u_i^2 = 1$, (ii) symmetric such that $\mathbb{E}u_i^3 = 0$, and (iii) its kurtosis is finite such that $\mathbb{E}u_i^4 = r_4 < \infty$.*

The location-scale family with Assumption 2.2 encompasses many variational families used in practice, such as Gaussians, Student-$t$ with a high-enough degrees of freedom $\nu$, Laplace, and so on, and enables the use of the reparametrization gradient.

While the choice of $\varphi$ gives control over the tail behavior of the family, the choice of the structure of the scale matrix $C$ gives control over how much correlation between coordinates of $\ell$ the variational approximation can represent. This ability to represent correlations is often referred to as the "expressiveness" of a variational family, where the most expressive choice is the following:

**Definition 2.3** (Full-Rank Location-Scale Family)**.** *We say $\mathcal{Q}$ is a full-rank location-scale family if it satisfies Definition 2.1 and, for any $C \in \mathcal{C}$, $C$ is invertible and the squared $Cs$, $CC^\top$, span the whole space of dense $\mathbb{R}^{d \times d}$ positive definite matrices as $\{CC^\top \mid C \in \mathcal{C}\} = \mathcal{S}_{\succ 0}^d$.*

Typically, full-rank location-scale families are formed by setting $\mathcal{C}$ to be the set of invertible triangular matrices (the "Cholesky factor parametrization"; Kucukelbir et al., 2017; Titsias and Lázaro-Gredilla, 2014) or the set of symmetric square roots (Domke, 2020; Domke et al., 2023). Adding further restrictions on $\mathcal{C}$ forms various subsets of the broader location-scale family. In this work, we focus on the case where $C \in \mathcal{C}$ is restricted to be diagonal such that $\mathcal{C} \subset \mathbb{D}^d$, which is known as the mean-field approximation (Hinton and van Camp, 1993; Peterson and Hartman, 1989):

**Definition 2.4** (Mean-Field Location-Scale Family)**.** *We say $\mathcal{Q}$ is a mean-field location-scale family if it satisfies Definition 2.1 and all $C \in \mathcal{C}$ are diagonal such that $\mathcal{C} \subset \mathbb{D}^d$ .*

## 2.3 Algorithm Setup

Recall that BBVI is essentially SGD in the space of parameters of the variational distribution. Therefore, we have to define the space of parameters. For this, we use the "linear" parametrization:

$$\Lambda = \left\{\lambda = (m, C) \mid m \in \mathbb{R}^d, C \in \mathbb{D}_{\succ 0}^d\right\} \subset \mathbb{R}^p \ . \tag{3}$$

Under this parametrization, the desirable properties of $\ell$ easily transfer to $f$. For instance, if $\ell$ is $\mu$-strongly convex and $L$-smooth, $f$ is also $\mu$-strongly convex and $L$-smooth (Domke, 2020). This contrasts with "non-linear parametrizations" commonly used in practice, such as making the diagonal positive by $C_{ii} = \exp(\lambda_{C_{ii}})$. Such practice rules out transfer of strong convexity and smoothness (Kim et al., 2023a) unless constraints such as $C_{ii} \geq \delta$ for some $\delta > 0$ are enforced (Hotti et al., 2024). (Though they can sometimes be beneficial by reducing gradient variance; Hotti et al., 2024; Kim et al., 2023b.) The flip side of using the linear parametrization is that we must now enforce the constraint $C \succ 0$. Furthermore, $h$ then becomes non-smooth with respect to $C$:

$$ h(\lambda) \quad = \quad -\mathbb{H}(q_\lambda) \quad = \quad -\log|\det C| - d\,\mathbb{H}(\varphi) \quad = \quad -\sum_{i=1}^{d} \log|C_{ii}| - d\,\mathbb{H}(\varphi) \,. \tag{4} $$

This corresponds to log-barrier functions (Parikh and Boyd, 2014, §6.7.5), which are non-smooth. Thus, the optimization algorithm must somehow deal with these difficulties (Domke, 2020).

In this work, we will rely on the proximal variant of stochastic gradient descent (SGD; Bottou, 1999; Bottou et al., 2018; Nemirovski et al., 2009; Robbins and Monro, 1951; Shalev-Shwartz et al., 2011), often referred to as stochastic proximal gradient descent (SPGD; Nemirovski et al., 2009). Proximal methods are a family of methods that rely on proximal operators (Parikh and Boyd, 2014), which are well defined as long as the following hold:

**Assumption 2.5.** $h : \Lambda \to \mathbb{R} \cup \{+\infty\}$ *is convex, bounded below, and lower semi-continuous.*

The non-smoothness of $h$ and the domain constraint are handled by the proximal operator

$$ \mathrm{prox}_{\gamma h}(\lambda) \triangleq \arg\min_{\lambda' \in \Lambda} \left\{ h(\lambda') + 1/(2\gamma)\|\lambda - \lambda'\|_2^2 \right\} \,, $$

while the intractability of $f$ is handled through stochastic estimates of $\nabla f$ (Definition 2.6). For a step size schedule $(\gamma_t)_{t \geq 0}$, $\widehat{\nabla f}$, an unbiased estimator of $\nabla f(\lambda_t) = \mathbb{E}\widehat{\nabla f}(\lambda_t; u)$, and a sequence of i.i.d. noise $(u_t)_{t \geq 0}$, for each $t \geq 0$, SPGD iterates

$$ \lambda_{t+1} = \mathrm{prox}_{\gamma_t h}\big(\lambda_t - \gamma_t \widehat{\nabla f}(\lambda; u_t)\big) \,. $$

In the case of BBVI with a mean-field location-scale family, the proximal operator of Eq. (4) is identical to that of log-barrier functions (Parikh and Boyd, 2014, §6.7.5):

$$ \mathrm{prox}_{\gamma h}(\lambda = (m, C)) = (m, C'), \quad \text{where} \quad C'_{ii} = (1/2)\Big(C_{ii} + \sqrt{C_{ii}^2 + 4\gamma}\Big) \,. $$

Instead of using SPGD, one can also use projected SGD, where $C$ is projected to a subset where $F$ is smooth (Domke, 2020) and use the "closed-form entropy" gradient $\widehat{\nabla F} \triangleq \widehat{\nabla f} + \nabla h$ (Kucukelbir et al., 2017; Titsias and Lázaro-Gredilla, 2014). However, the resulting theoretical guarantees are indistinguishable (Domke et al., 2023), and the need for setting a closed domain of $C$ is inconvenient. Therefore, we only consider SPGD. But our results can easily be applied to projected SGD.

For $\widehat{\nabla f}$, we will use the classic *reparametrization gradient* (Ho and Cao, 1983; Rubinstein, 1992):

**Definition 2.6** (Reparametrization Gradient). *For a differentiable function $\ell : \mathbb{R}^d \to \mathbb{R}$,*

$$ \widehat{\nabla f}(\lambda; u) \triangleq \nabla_\lambda \ell\left(\mathcal{T}_\lambda\left(u\right)\right) = \frac{\partial \mathcal{T}_\lambda(u)}{\partial \lambda} \nabla \ell\left(\mathcal{T}_\lambda\left(u\right)\right) \,, \quad \text{where} \quad u \sim \varphi \,, $$

*is an unbiased estimator of $\nabla f$ such that $\nabla_\lambda \mathbb{E}_{z \sim q_\lambda} \ell(z) = \nabla f(\lambda)$.*

The reparametrization gradient, also known as the push-in gradient or pathwise gradient, was introduced to VI by Kingma and Welling (2014); Rezende et al. (2014); Titsias and Lázaro-Gredilla (2014). (See also the reviews by Glasserman 1991; Mohamed et al. 2020; Pflug 1996.) It is empirically observed to outperform alternatives (Kucukelbir et al., 2017; Mohamed et al., 2020) such as the score gradient (Glynn, 1990; Williams, 1992) and *de facto* standard whenever $\ell$ is differentiable. (Though theoretical evidence of this superiority is limited to the quadratic setting; Xu et al., 2019.)

## 2.4 General Analysis of Stochastic Proximal Gradient Descent

Analyzing the convergence of BBVI corresponds to analyzing the convergence of SPGD (or more broadly, of SGD) for the class of problems that corresponds to BBVI. For this, we will first discuss sufficient conditions for the convergence of SPGD and the resulting consequences.

**Assumption 2.7** (Lipschitz Gradients in Expectation). *There exists some constant $\mathcal{L} \in [0, \infty)$ such that, for all $\lambda, \lambda' \in \Lambda$,*

$$ \mathbb{E}\|\widehat{\nabla f}(\lambda; u) - \widehat{\nabla f}(\lambda'; u)\|_2^2 \leq \mathcal{L}^2 \|\lambda - \lambda'\|_2^2 \,. $$

**Assumption 2.8** (Bounded Variance). *There exists some constant $\sigma \in [0, \infty)$ such that, for all $\lambda_* \in \arg\min_{\lambda \in \Lambda} F(\lambda)$,*

$$\mathbb{E}\|\widehat{\nabla}f(\lambda_*; u)\|_2^2 \leq \sigma^2 .$$

Both assumptions were initially used by Bach and Moulines (2011, Assumptions H2 and H4) to analyze the convergence of SGD. Here, Assumption 2.7 serves as an analog of $L$-smoothness, and thus determines the largest stepsize we can use. The strategy of combining Assumptions 2.7 and 2.8 is referred to as "variance transfer" (Garrigos and Gower, 2023, §4.3.3). Previously, for analyzing BBVI, a slightly different assumption called quadratically-bounded variance (QV)—which assumes the existence of $\alpha, \beta \in [0, +\infty)$ such that, for all $\lambda \in \Lambda$, $\mathbb{E}\|\widehat{\nabla}f(\lambda; u)\|_2^2 \leq \alpha\|\lambda - \lambda_*\|_2^2 + \beta$ holds— has been commonly used (Domke, 2019; Domke et al., 2023; Kim et al., 2024b). While similar, our assumptions result in a constant-factor improvement in the resulting bounds.

For the analysis, we will use a two-stage step size schedule (Gower et al., 2019, Theorem 3.2):

$$\gamma_t = \begin{cases} \gamma_0 & \text{if } t \leq t_* \\ \frac{1}{\mu}\frac{2(t+\tau)+1}{(t+\tau+1)^2} & \text{if } t \geq t_* + 1 \end{cases} , \quad \text{where} \quad 0 < \gamma_0 \leq \frac{\mu}{2\mathcal{L}^2} \tag{5}$$

This operates by first maintaining a fixed step size $\gamma_0$ until some switching time $t_* \in \{0, \ldots, T\}$, and then switches to the $1/t$ schedule of Lacoste-Julien et al. (2012) with an offset $\tau \geq 0$.

Under Assumptions 2.7 and 2.8, we can now provide a complexity guarantee for solving Eq. (1) via SPGD. Since BBVI consists of a subset of Eq. (1), establishing Assumptions 2.7 and 2.8 and invoking the following result will constitute our complexity guarantee for BBVI.

**Proposition 2.9.** *Suppose $f$ is $\mu$-strongly convex, $h$ satisfies Assumption 2.5, and $\widehat{\nabla}f$ satisfies Assumptions 2.7 and 2.8. Then, for the global optimum $\lambda_* = \arg\min_{\lambda \in \Lambda} F(\lambda)$, $\Delta \triangleq \|\lambda_0 - \lambda_*\|_2$, some $t_*, \tau$, and $\gamma_0$ (explicit in the proof), SPGD with the step size schedule in Eq. (5) guarantees*

$$T \geq O\left\{ \frac{\sigma^2}{\mu^2}\frac{1}{\epsilon} + \frac{\sigma\mathcal{L}}{\mu^2}\log\left(\frac{\mathcal{L}^2}{\sigma^2}\Delta^2\right)\frac{1}{\sqrt{\epsilon}} + \frac{\mathcal{L}^2}{\mu^2}\log\left(\Delta^2\frac{1}{\epsilon}\right) + 1 \right\} \quad \Rightarrow \quad \mathbb{E}\|\lambda_T - \lambda_*\|_2^2 \leq \epsilon .$$

*Proof.* See the *full proof* in Appendix B.1.1, p. 19. $\qquad\square$

This result is a slight improvement over past analysis of SPGD with Eq. (5) (Domke et al., 2023, Theorem 7). In particular, the dependence on the initialization $\Delta$ has been improved to be logarithmic instead of polynomial. Furthermore, it encompasses the case where we have "interpolation" ($\sigma^2 = 0$; Kim et al., 2024b; Schmidt and Roux, 2013; Vaswani et al., 2019) automatically resulting in a $O(\log 1/\epsilon)$ complexity. A similar result for vanilla SGD, where the dependence on both $\epsilon$ and $\Delta$ is optimized simultaneously, was reported by Stich (2019). However, this result required a schedule that depends on the maximum number of iterations $T$. Compared with this, our result provides an *any-time* guarantee that holds for any number of iterations.

For a non-strongly convex $f$, using the strategy of Domke et al. (2023, Theorem 8 and 11) should yield a corresponding $O(1/\epsilon^2)$ complexity guarantee under the same set of assumptions. However, this requires fixing the horizon $T$ in advance, and it is currently unknown how to obtain an anytime $O(1/\sqrt{T})$ convergence bound for SGD under Assumptions 2.7 and 2.8 or QV. If one moves away from the canonical SGD update by incorporating Halpern iterations (Halpern, 1967), it is possible to obtain any-time convergence under a QV-like assumption (Alacaoglu et al., 2025).

## 3 Main Results

### 3.1 General Result

For our results, we impose an additional assumption that is a generalization of $L$-smoothness under twice differentiability of $\ell$.

**Assumption 3.1.** *$\ell$ is twice differentiable and, for all $z \in \mathbb{R}^d$, there exist some matrix $H \in \mathbb{R}^{d \times d}$ and constant $\delta \in [0, \infty)$ satisfying*

$$\|H\|_2 < \infty \quad \text{and} \quad \|\nabla^2\ell(z) - H\|_2 \leq \delta .$$

Notably, if $\ell$ is twice differentiable, $\mu$-strongly convex, $L$-smooth, it already satisfies Assumption 3.1 with $H = \frac{L+\mu}{2}I_d$ and $\delta = \frac{L-\mu}{2}$. If $\ell$ is only $L$-smooth, it satisfies it with $H = 0_{d \times d}$ and $\delta = L$. The key advantage of this assumption, however, is that it characterizes Hessians that are not necessarily well-conditioned, but almost constant. This crucially affects the dimension dependence.

Given our assumptions on the target function $\ell$, variational family $\mathcal{Q}$, and our choice of gradient estimator, we can guarantee that SPGD applied to a problem structure corresponding to BBVI (Eq. (1)) achieves a given level of accuracy $\epsilon$ after $\mathrm{O}(g(d, H, \delta, \mu, \varphi)\epsilon^{-1})$ number of iterations:

**Theorem 3.2.** *Suppose the following hold:*

1. *$\ell$ is $\mu$-strongly convex and satisfies Assumption 3.1 and $\mu \leq \sigma_{\min}(H) \leq \sigma_{\max}(H) \leq L$.*

2. *$h$ satisfies Assumption 2.5.*

3. *$\mathcal{Q}$ is a mean-field location-scale family, where Assumption 2.2 holds.*

4. *$\widehat{\nabla f}$ is the reparametrization gradient.*

*Denote the global optimum $\lambda_* = (m_*, C_*) = \arg\min_{\lambda \in \Lambda} F(\lambda)$, the irreducible gradient noise as $\sigma_*^2 \triangleq \|m_* - \bar{z}\|_2^2 + \|C_*\|_{\mathrm{F}}^2$, and the stationary point of $\ell$ as $\bar{z} \triangleq \arg\min_{z \in \mathbb{R}^d} \ell(z)$. Then, for some $t_*$, $\tau$, and $\gamma_0$ (explicit in the proof), SPGD with the step size schedule in Eq. (5) guarantees*

$$T \geq \mathrm{O}\Big\{g(d, H, \delta, \mu, \varphi)\Big(\sigma_*^2\epsilon^{-1} + \sigma_* \log\big(\|\lambda_0 - \lambda_*\|_2^2\big)\epsilon^{-1/2}\Big)\Big\} \quad \Rightarrow \quad \mathbb{E}\|\lambda_T - \lambda_*\|_2^2 \leq \epsilon \,,$$

*where*

$$g(d, H, \delta, \mu, \varphi) \triangleq 2\left(1 + r_4\right)\big(\|H\|_2^2/\mu^2\big) + 4\big(\delta^2/\mu^2\big)\Big((1/2) + r_4 + \mathbb{E}\max_{j=1,\ldots,d} u_j^2\Big) \,.$$

*Proof.* The *full proof* can be found in Appendix B.2.1, p. 25. □

Due to the identity $\|\lambda - \lambda'\|_2^2 = \mathbb{E}_{u \sim \varphi^{\otimes d}}\|\mathcal{T}_\lambda(u) - \mathcal{T}_{\lambda'}(u)\|_2^2$ (Lemma A.3), which is the squared cost of a coupling between $q_{\lambda_T}$ and $q_{\lambda_*}$, our guarantee also translates to a guarantee in Wasserstein-2 distance: $\mathbb{E}\|\lambda_T - \lambda_*\|_2^2 \leq \epsilon \Rightarrow \mathbb{E}W_2(q_{\lambda_T}, q_{\lambda_*})^2 \leq \epsilon$. In the general case where $\delta > 0$, the dimension dependence enters through $\mathbb{E}\max_{j=1,\ldots,d} u_j^2$, which depends on the order-statistics of the base distribution $\varphi$. In case $\ell$ is a quadratic, corresponding to $\pi$ being a Gaussian target distribution in the BBVI context, there exists some $H$ such that $\nabla^2\ell(z) = H$ for all $z \in \mathbb{R}^d$. Thus, Assumption 3.1 holds with $\delta = 0$, implying a dimension-independent convergence rate. We will present additional special cases with more explicit choices of $\varphi$ in the next section.

In case we do not want to assume Assumption 3.1 and only assume that $\ell$ is $\mu$-strongly convex and $L$-smooth instead, we can replace them with the generic choices of $H = \frac{L+\mu}{2}\mathrm{I}_d$ and $\delta = \frac{L-\mu}{2}$, which hold for all $\ell$s that are $\mu$-strongly convex, $L$-smooth, and twice differentiable. This then makes the role of the condition number $\kappa \triangleq L/\mu$ more explicit.

**Corollary 3.3.** *Suppose $\ell$ is is twice differentiable, $\mu$-strongly convex, and $L$-smooth. Then, denoting the condition number as $\kappa \triangleq L/\mu$, Theorem 3.2 holds with*

$$g\Big(d, \tfrac{L+\mu}{2}\mathrm{I}_d, \tfrac{L-\mu}{2}, \mu, \varphi\Big) = (1/2)(1 + r_4)(\kappa + 1)^2 + (\kappa - 1)^2\Big((1/2) + r_4 + \mathbb{E}\max_{j=1,\ldots,d} u_j^2\Big) \,.$$

This makes the $\mathrm{O}(\kappa^2)$ condition number dependence explicit, but the downside is that we lose dimension independence in the case of ill-conditioned quadratic $\ell$s. This fact suggests that dimension dependence is more fundamentally related to how close the Hessian is to a constant rather than how well-conditioned it is.

## 3.2 Special Cases with Benign Dimension Dependence

We now present some special cases of Theorem 3.2, which has yet to exhibit an explicit dependence on dimensionality. As mentioned in the previous section, dimension dependence depends on the order statistics of $\varphi$, which is related to the tail behavior of $\varphi$.

**Variational Families with Sub-Gaussian Tails.** The most commonly used variational family in practice is the Gaussian variational family. More broadly, for sub-Gaussian variational families, $u_i^2$ is sub-exponential and therefore admits a moment generating function (MGF) (Wainwright, 2019, Theorem 2.6), which leads to a $\mathrm{O}(\log d)$ explicit dimension dependence.

**Proposition 3.4.** *Suppose there exists some $t > 0$ such that the MGF of $u_i^2$ satisfies $M_{u_i^2}(t) < \infty$. Then*

$$\mathbb{E}\max_{i=1,\ldots,d} u_i^2 \quad \leq \quad (1/t)\left(\log M_{u_i^2}(t) + \log d\right) .$$

*For example, if $\varphi$ is a standard Gaussian, then*

$$g(d, H, \delta, \mu, \varphi) \quad \leq \quad 8\big(\|H\|_2^2/\mu^2\big) + \big(\delta^2/\mu^2\big)(22 + 16\log d) .$$

*Proof.* The *full proof* can be found in Appendix B.2.2, p. 26. □

**Variational Families with Finite Higher Moments.** For families with tails heavier than sub-Gaussian, however, $u_i^2$ may not have an MGF. While we then lose the $\mathrm{O}(\log d)$ dependence, we may still obtain a polynomial dependence that can be better than $\mathrm{O}(\sqrt{d})$ obtained in previous works (Kim et al., 2023b). In particular, the result that will follow states that the highest order of the available moments determines the order of dimension dependence. For Student-$t$ families, this implies that using a high-enough degree of freedom $\nu$ can make the dimension dependence benign.

**Proposition 3.5.** *Suppose, for $k \geq 2$, the $k$th moment of $u_i^2$ is finite as $r_{2k} = \mathbb{E}u_i^{2k} < \infty$. Then*
$$\mathbb{E}\max_{i=1,\ldots,d} u_i^2 \quad \leq \quad \sqrt{2}\, d^{1/k}\, r_{2k}^{1/k}\,.$$
*For example, if $\varphi$ is a Student-$t$ with $\nu > 4$ degrees of freedom and unit variance, then*
$$g(d, H, \delta, \mu, \varphi) \quad \leq \quad 8(\|H\|_2^2/\mu^2) + (\delta^2/\mu^2)\left(16 + \sqrt{2}\,\nu^3 d^{\frac{2}{\nu-2}}\right)\,.$$

*Proof.* See the *full proof* in Appendix B.2.3, p. 28. $\qquad\qquad\qquad\qquad\qquad\qquad\qquad\square$

# 4 Analysis of Gradient Variance

## 4.1 Overview

The key technical contribution of this work is analyzing the gradient variance and thus establishing the constants $\mathcal{L}$ (Assumption 2.7) and $\sigma^2$ (Assumption 2.8), which boils down to analyzing
$$\mathbb{E}\|\widehat{\nabla f}(\lambda; u) - \widehat{\nabla f}(\lambda'; u)\|_2^2 = \mathbb{E}\left\|\frac{\partial \mathcal{T}_\lambda(u)}{\partial \lambda}\nabla \ell(\mathcal{T}_\lambda(u)) - \frac{\partial \mathcal{T}_{\lambda'}(u)}{\partial \lambda'}\nabla \ell(\mathcal{T}_{\lambda'}(u))\right\|_2^2$$
$$= \mathbb{E}\left\|\frac{\partial \mathcal{T}_\lambda(u)}{\partial \lambda}(\nabla \ell(\mathcal{T}_\lambda(u)) - \nabla \ell(\mathcal{T}_{\lambda'}(u)))\right\|_2^2\,,$$
where the equality follows from the fact that the Jacobian $\partial \mathcal{T}_\lambda(u)/\partial \lambda$ does not depend on $\lambda$. For mean-field location-scale variational families, the squared Jacobian follows as
$$\left(\frac{\partial \mathcal{T}_\lambda(u)}{\partial \lambda}\right)^\top \frac{\partial \mathcal{T}_\lambda(u)}{\partial \lambda} = \mathrm{I}_d + U^2\,,$$
where $U \triangleq \mathrm{diag}(u_1, \ldots, u_d)$ (Kim et al., 2023b). This implies that
$$\mathbb{E}\|\widehat{\nabla f}(\lambda; u) - \widehat{\nabla f}(\lambda'; u)\|_2^2$$
$$= \underbrace{\mathbb{E}\|\nabla \ell(\mathcal{T}_\lambda(u)) - \nabla \ell(\mathcal{T}_{\lambda'}(u))\|_2^2}_{\triangleq V_{\mathrm{loc}}} + \underbrace{\mathbb{E}\|U(\nabla \ell(\mathcal{T}_\lambda(u)) - \nabla \ell(\mathcal{T}_{\lambda'}(u)))\|_2^2}_{\triangleq V_{\mathrm{scale}}}\,. \quad (6)$$
Our goal is to bound each term by $\|\lambda - \lambda'\|_2^2$.

In order to solve the expectations, we need to simplify the $\nabla \ell$ terms. For instance, for the gradient of the location $V_{\mathrm{loc}}$, assuming that $\ell$ is $L$-smooth allows for a quadratic approximation. That is,
$$V_{\mathrm{loc}} = \mathbb{E}\|\nabla \ell(\mathcal{T}_\lambda(u)) - \nabla \ell(\mathcal{T}_{\lambda'}(u))\|_2^2 \leq L^2 \mathbb{E}\|\mathcal{T}_\lambda(u) - \mathcal{T}_{\lambda'}(u)\|_2^2 = L^2\|\lambda - \lambda'\|_2^2\,,$$
where the last equality is by Lemma A.3.

Now, it is tempting to use the same quadratic approximation strategy for the gradient of the scale $V_{\mathrm{scale}}$. Indeed, this strategy was used by Domke (2019) to bound the gradient variance of full-rank location-scale variational families and by Ko et al. (2024) for structured location-scale variational families. Unfortunately, this strategy does not immediately apply to mean-field families due to the matrix $U$. We somehow have to decouple $\nabla \ell(\mathcal{T}_\lambda(u)) - \nabla \ell(\mathcal{T}_{\lambda'}(u))$ and $U$, but in a way that does not lose the correlation between the two; the correlation leads to cancellations critical to obtaining a tight bound. Kim et al. (2023b) used the inequality
$$V_{\mathrm{scale}} \leq \mathbb{E}\|U^2\|_{\mathrm{F}}\|\nabla \ell(\mathcal{T}_\lambda(u)) - \nabla \ell(\mathcal{T}_{\lambda'}(u))\|_2^2\,, \quad (7)$$
which resulted in a dimension dependence of $\mathrm{O}(r_4\sqrt{d})$ after solving the expectation. The key question is whether this dimension dependence can be improved. Due to the ordering of norms $\|\cdot\|_2 \leq \|\cdot\|_{\mathrm{F}}$, it is natural to consider the tighter inequality
$$V_{\mathrm{scale}} \leq \mathbb{E}\|U\|_2^2\|\nabla \ell(\mathcal{T}_\lambda(u)) - \nabla \ell(\mathcal{T}_{\lambda'}(u))\|_2^2\,.$$
(This step corresponds to Eq. (8) in the proof sketch of the upcoming result.) The main challenge, however, is solving the resulting expectation in a way that is also tight with respect to $d$. We will see that this requires a careful probabilistic analysis.

## 4.2 Upper Bound on Gradient Variance

We now formally state our upper bound on the gradient variance. In the context of proving Theorem 3.2, the following lemma implies both Assumption 2.7 and Assumption 2.2. (See the proof of Theorem 3.2.) We provide a corresponding unimprovability result in Section 4.3.

**Lemma 4.1.** *Suppose Assumptions 2.2 and 3.1 hold, $\mathcal{Q}$ is a mean-field location-family, and $\widehat{\nabla} f$ is the reparametrization gradient. Then, for any $\lambda, \lambda' \in \mathbb{R}^d \times \mathbb{D}^d$.*

$$\mathbb{E} \| \widehat{\nabla} f(\lambda; \boldsymbol{u}) - \widehat{\nabla} f(\lambda'; \boldsymbol{u}) \|_2^2 \leq \left\{ 2(1 + r_4) \| H \|_2^2 + 4 \delta^2 \left( 1/2 + r_4 + \mathbb{E} \max_{j=1,\ldots,d} u_j^2 \right) \right\} \| \lambda - \lambda' \|_2^2 \,.$$

*Proof Sketch.* For the proof sketch, we will assume that $\ell$ is $L$-smooth instead of taking Assumption 3.1. This will vastly simplify the analysis and let us focus on the key elements.

Recall $V_{\text{scale}}$ in Eq. (6). Applying the operator norm and the $L$-smoothness of $\ell$ yields

$$V_{\text{scale}} \quad \leq \quad \mathbb{E} \| \boldsymbol{U} \|_2^2 \| \nabla \ell(\mathcal{T}_\lambda(\boldsymbol{u})) - \nabla \ell(\mathcal{T}_{\lambda'}(\boldsymbol{u})) \|_2^2 \quad \leq \quad L^2 \mathbb{E} \| \boldsymbol{U} \|_2^2 \| \mathcal{T}_\lambda(\boldsymbol{u}) - \mathcal{T}_{\lambda'}(\boldsymbol{u}) \|_2^2 \,. \quad (8)$$

It remains to solve the expectation over $u_1, \ldots, u_d$. Denote

$$\lambda = (m, C), \qquad \lambda' = (m', C'), \qquad \bar{m} \triangleq m - m', \quad \text{and} \quad \bar{C} \triangleq C - C' \,,$$

recall that $\mathcal{T}_\lambda(u) = Cu + m$ (Definition 2.1), and notice that, since $\boldsymbol{U}$ is a diagonal matrix, $\| \boldsymbol{U} \|_2^2 = \max_{i=1,\ldots,d} u_j^2$. Then we can rewrite Eq. (8) as

$$\begin{aligned} V_{\text{scale}} \quad &\leq \quad L^2 \mathbb{E} \left( \max_{i=1,\ldots,d} u_j^2 \right) \textstyle\sum_{i=1}^d (C_{ii} u_i + m_i - C'_{ii} u_i - m'_i)^2 \\ &= \quad L^2 \mathbb{E} \left( \max_{i=1,\ldots,d} u_j^2 \right) \textstyle\sum_{i=1}^d \left( \bar{C}_{ii} u_i + \bar{m}_i \right)^2 \\ &\leq \quad L^2 \mathbb{E} \left( \max_{i=1,\ldots,d} u_j^2 \right) \textstyle\sum_{i=1}^d \left( 2 \bar{C}_{ii}^2 u_i^2 + 2 \bar{m}_i^2 \right) \,. \qquad \text{(Young's inequality)} \end{aligned}$$

The problematic term is

$$\mathbb{E} \left( \max_{j=1,\ldots,d} u_j^2 \right) \textstyle\sum_{i=1}^d \bar{C}_{ii}^2 u_i^2 \quad = \quad \mathbb{E} u_{i_*}^2 \textstyle\sum_{i=1}^d \bar{C}_{ii}^2 u_i^2 \quad = \quad \mathbb{E} \left[ \bar{C}_{i_* i_*}^2 u_{i_*}^4 + \textstyle\sum_{j \neq i_*}^d \bar{C}_{jj}^2 u_{i_*}^2 u_j^2 \right] \,,$$

where $i_* = \arg\max_{i=1,\ldots,d} u_i^2$ is the coordinate of maximum magnitude. Here, $u_{i_*}^4 = \max_{i=1,\ldots,d} u_i^4$ is a heavy-tailed quantity that generally grows fast in $d$, unlike $u_{i_*}^2$. (*e.g.*, for a Gaussian $u_i$, $u_{i_*}^2$ has an MGF but $u_{i_*}^4$ does not.) Therefore, a benign dimension dependence might appear futile. Notice, however, that the problematic term only affects a single dimension: the coordinate indicated by $i_*$. A probabilistic analysis reveals that as $d$ increases, the effect of $u_{i_*}^4$ becomes averaged out and the effect of the remaining term involving $u_i^2$ dominates. More formally,

$$\mathbb{E} u_{i_*}^2 \textstyle\sum_{i=1}^d \bar{C}_{ii}^2 u_i^2 \quad = \quad \textstyle\sum_{i=1}^d \bar{C}_{ii}^2 \, \mathbb{E} \left[ u_{i_*}^4 \mathbb{1}\{ i_* = i \} + u_{i_*}^2 u_i^2 \mathbb{1}\{ i_* \neq i \} \right] \,,$$

where

$$\mathbb{E} \left[ u_{i_*}^4 \mathbb{1}\{ i_* = i \} \right] \quad = \quad \mathbb{E} \left[ u_{i_*}^4 \right] \mathbb{E} [ \mathbb{1}\{ i_* = i \} ] \quad = \quad \mathbb{E} \left[ u_{i_*}^4 \right] \mathbb{P}[ i_* = i ] \quad = \quad \mathbb{E} \left[ u_{i_*}^4 \right] (1/d) \,.$$

Since the maximum of $d$ random variable is always smaller than their sum, the probability of the maximally random event, $\mathbb{P}[i_* = i] = 1/d$, kills off the dimensional growth of $u_{i_*}^4$. In fact, using the crude bound $\mathbb{E} u_{i_*}^4 \leq \mathbb{E} \sum_{i=1}^d u_i^4 = d r_4$, where the last equality is due to Assumption 2.2, is enough to make this term independent of $d$. The remaining dimension dependence comes from $u_{i_*}^2$:

$$\mathbb{E} \left[ u_{i_*}^2 u_i^2 \mathbb{1}\{ i_* \neq i \} \right] \quad = \quad \mathbb{E} \left[ \max_{j \neq i} u_j^2 u_i^2 \mathbb{1}\{ i_* \neq i \} \right] \quad \leq \quad \mathbb{E} \left[ \max_{j=1,\ldots,d-1} u_j^2 \right] \mathbb{E} [ u_i^2 ] \quad = \quad \mathbb{E} \max_{j=1,\ldots,d-1} u_j^2 \,,$$

where the last equality follows from Assumption 2.2. Therefore, we finally obtain

$$\begin{aligned} V_{\text{scale}} &\leq 2 L^2 \sum_{i=1}^d \left[ \left( \mathbb{E} \max_{j=1,\ldots,d-1} u_j^2 + r_4 \right) \bar{C}_{ii}^2 + \mathbb{E} \max_{j=1,\ldots,d} u_j^2 \bar{m}_i^2 \right] \\ &\leq 2 L^2 \left( \mathbb{E} \max_{j=1,\ldots,d} u_j^2 + r_4 \right) \left( \| \bar{m} \|_2^2 + \| \bar{C} \|_{\mathrm{F}}^2 \right) \\ &= 2 L^2 \left( \mathbb{E} \max_{j=1,\ldots,d} u_j^2 + r_4 \right) \| \lambda - \lambda' \|_2^2 \,. \end{aligned}$$

The full proof performs an analogous analysis under the more general Assumption 3.1. $\qquad\square$

See the *full proof* in Appendix B.3.1, p. 30.

### 4.3 Unimprovability

We also demonstrate a lower bound, which implies that Lemma 4.1 cannot be improved when relying on the spectral bounds on $\nabla^2 \ell$. From Eq. (6) and the fundamental theorem of calculus,

$$\mathbb{E}\|\widehat{\nabla f}(\lambda; u)\|_2^2 \;\geq\; \mathbb{E}\|U(\nabla \ell(z) - \nabla \ell(\bar{z}))\|_2^2 \;=\; \mathbb{E}\big\|U \int_0^1 \nabla^2 \ell(z^w)(z - \bar{z}) \mathrm{d}w\big\|_2^2, \quad (9)$$

where $\bar{z} \in \{z \mid \nabla \ell(z) = 0\}$ is a stationary point of $\ell$, $z \triangleq \mathcal{T}_\lambda(u)$, and $z^w \triangleq wz + (1-w)\bar{z}$. There exists a matrix-valued function with bounded singular values that lower-bounds this quantity:

**Proposition 4.2.** *Suppose Assumption 2.2 holds and $\mathcal{Q}$ is a mean-field location-scale family. Then, for any $t > 0$, $d > 0$, $\mu, L \in (0, +\infty)$ satisfying $\mu \leq L$, there exists a matrix-valued function $H(z) : \mathbb{R}^d \to \mathbb{S}_{\succ 0}^d$ satisfying $\mu \mathrm{I}_d \preceq H \preceq L \mathrm{I}_d$ almost surely and a set of parameters $\lambda = (m, C) \in \mathbb{R}^d \times \mathbb{D}_{\succ 0}^d$ such that*

$$\mathbb{E}\big\|U \int_0^1 H(z^w)(z - \bar{z}) \mathrm{d}w\big\|_2^2 \geq \left\{ \frac{(L-\mu)^2}{4} - \frac{L^2}{2} \frac{\mathbb{E}\max_{i=1,\dots,d} u_i^4}{d} \right\} c(t, \varphi) \left\{ \mathbb{E}\max_{i=1,\dots,d-1} u_i^2 - t \right\} \|C\|_{\mathrm{F}}^2 \;.$$

*where $c(t, \varphi) > 0$ is a constant only dependent on $t$ and $\varphi$.*

*Proof.* The *full proof* can be found in Appendix B.3.4, p. 36. $\qquad\square$

For Gaussians, $\mathbb{E}\max_i u_i^4$ is upper bounded as $\mathrm{O}(\sqrt{d})$ (Gumbel, 1954, Eq. 1.6), which means the negative term vanishes at a $1/\sqrt{d}$ rate. Furthermore, $\mathbb{E}\max_i u_i^2 \geq (\mathbb{E}\max_i u_i)^2 = \Omega(\log d)$ by the well-known lower bound on the expected maximum of i.i.d. Gaussians (Wainwright, 2019, Exercise 2.11.(b)). Combining these facts with Proposition 4.2 yield a $\Omega(L^2 \log d)$ bound on Eq. (9).

**Remark 4.3.** *It is not obvious that the rows of our worst-case example $H_{\mathrm{worst}}$ form conservative vector fields. This means that Proposition 4.2 does not assert the existence of a function $\ell$ that satisfies $\nabla^2 \ell = H_{\mathrm{worst}}$. However, it does suggest that one cannot improve Lemma 4.1 by relying only on spectral bounds on the Hessian.*

## 5 Discussion

### 5.1 Related Works

Early results analyzing VI had to rely on assumptions that either: (i) do not hold on Gaussian targets, (ii) are difficult to verify, or (iii) require bounds on the domain (Alquier and Ridgway, 2020; Buchholz et al., 2018; Fan et al., 2015; Fujisawa and Sato, 2021; Khan et al., 2016; Liu and Owen, 2021; Nguyen et al., 2025; Regier et al., 2017). coordinate-ascent VI (CAVI), in particular, was studied on specific models (Ghorbani et al., 2019; Zhang and Zhou, 2020) only. Under general and verifiable assumptions, Xu and Campbell (2022) obtained asymptotic convergence guarantees, while partial results, such as bounds on the gradient variance (Domke, 2019; Fan et al., 2015; Kim et al., 2023b), or regularity of the ELBO (Challis and Barber, 2013; Domke, 2020; Titsias and Lázaro-Gredilla, 2014), were known.

It was only recently that non-asymptotic quantitative convergence under realizable and verifiable assumptions was established. For BBVI specifically, Hoffman and Ma (2020) first proved convergence on Gaussian targets (quadratic $\ell$), while Domke et al. (2023); Kim et al. (2023a, 2024b) proved the first results on strongly convex and smooth functions with location-scale families. Surendran et al. (2025) extended these results to non-convex smooth functions and more complex variational family parametrizations, and Cheng et al. (2024) analyzed a variant of semi-implicit VI. The results of Domke et al.; Hoffman and Ma; Kim et al., who focused on full-rank families, suggest a $\mathrm{O}(d)$ dimension dependence in the iteration complexity. On the other hand, Kim et al. (2023a) reported a $\mathrm{O}(\sqrt{d})$ dimension dependence for mean-field location-scale families, while conjecturing $\mathrm{O}(\log d)$ dependence, based on the partial result of Kim et al. (2023b). For targets with a diagonal Hessian structure, Ko et al. (2024, Corollary 1) show that mean-field families are dimension-independent.

Apart from BBVI, Wasserstein VI algorithms—which minimize the KL divergence on the Wasserstein geometry—provide non-asymptotic convergence guarantees. In particular, the algorithms by Diao et al. (2023); Lambert et al. (2022) optimize over the full-rank Gaussian family, while that of Jiang et al. (2025) optimizes over all mean-field families with bounded second moments. To guarantee $\mathbb{E}\mathrm{W}_2(q_{\lambda_T}, q_{\lambda_*})^2 \leq \epsilon$ on strongly log-concave and log-smooth targets, they all report an iteration complexity of $\mathrm{O}(d\epsilon^{-1} \log \epsilon^{-1})$. Meanwhile, under the same conditions, Arnese and Lacker

(2024); Lavenant and Zanella (2024) analyzed (block) CAVI, and reported an iteration complexity of $\mathrm{O}(d \log \epsilon^{-1})$. Bhattacharya et al. (2025) provides a concurrent result on CAVI, but relies on an assumption that departs from log-concavity and smoothness. Finally, Bhatia et al. (2022) analyzes a specialized algorithm optimizing over only the scale of Gaussians, which has a gradient query complexity of $\mathrm{O}(dk\epsilon^{-3})$, where $k$ is the user-chosen number of rank-1 factors in the scale matrix.

## 5.2 Conclusion

In this work, we proved that BBVI with mean-field location-scale families is able to converge with an iteration complexity with only a $\mathrm{O}(\log d)$ dimension dependence, as long as the tails of the family are sub-Gaussian. For high-dimensional targets, this suggests a substantial speed advantage over BBVI with full-rank families. In practice, the mean-field approximation can be combined with other design elements such as control variates (Boustati et al., 2020; Geffner and Domke, 2018, 2020; Miller et al., 2017; Roeder et al., 2017; Wang et al., 2024) and data-point subsampling (Kucukelbir et al., 2017; Titsias and Lázaro-Gredilla, 2014). Our analysis strategy should easily be combined with existing analyses (Kim et al., 2024a,b) for such design elements.

For a target distribution $\pi$ with a condition number of $\kappa$ and a target accuracy level $\epsilon$, we now know how to improve the dependence on $d$ and $\epsilon$ in the iteration complexity: Using less-expressive families such as mean-field (Theorem 3.2) or structured (Ko et al., 2024) families improves the dependence on $d$, while applying control variates to gradient estimators (Kim et al., 2024b) improves the dependence on $\epsilon$. However, it is currently unclear whether the dependence on $\kappa$ is tight or improvable. If it is tight, it would be worth investigating whether this can be provably improved through algorithmic modifications, for example, via stochastic second-order optimization methods (Byrd et al., 2016; Fan et al., 2015; Liu and Owen, 2021; Meng et al., 2020; Regier et al., 2017).

Another future direction would be to develop methods that are able to adaptively adjust the computational cost between $\mathrm{O}(\log d)$ and $\mathrm{O}(d)$ by trading statistical accuracy akin to the method of Bhatia et al. (2022). Existing BBVI schemes with "low-rank(-plus-diagonal)" families (Ong et al., 2018; Rezende et al., 2014; Tomczak et al., 2020) result in a non-smooth, non-Lipschitz, and non-convex landscape. This not only rules out typical theoretical convergence guarantees but also exhibits unstable and slow convergence in practice (Modi et al., 2025). Furthermore, understanding the statistical side of this trade-off will be an important direction. As of now, our understanding is restricted to either mean-field or full-rank families (Katsevich and Rigollet, 2024; Margossian and Saul, 2023, 2025; Wang and Blei, 2019a,b; Yang et al., 2020; Zhang and Gao, 2020) with little in between except for the work of Bhatia et al. (2022).

## Acknowledgments and Disclosure of Funding

The authors thank Anton Xue for helpful discussions and the reviewers for helpful suggestions.

K. Kim, J. R. Gardner were supported through the NSF award [IIS2145644]; Y.-A. Ma was supported by the NSF Award CCF-2112665 (TILOS), the DARPA AIE program, and the CDC-RFA-FT-23-0069; T. Campbell was supported by the NSERC Discovery Grant RGPIN-2025-04208.

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

# Contents

# A Auxiliary Lemmas

**Lemma A.1.** *Suppose Assumption 2.2 holds. Then $r_4 = \mathbb{E}u_i^4 \geq 1$.*

*Proof.* By Jensen's inequality $\mathbb{E}u_i^4 \geq \left(\mathbb{E}u_i^2\right)^2$. Lastly, $\mathbb{E}u_i^4 \geq \left(\mathbb{E}u_i^2\right)^2 = 1$ by Assumption 2.2. $\square$

**Lemma A.2.** *Suppose Assumption 2.2 holds and denote $U = \mathrm{diag}(u_1, \ldots, u_d)$. Then we have the following identities: (i) $\mathbb{E}uu^\top = I_d$, (ii) $\mathbb{E}U^2 = I_d$.*

*Proof.* From Assumption 2.2, we know that $\mathbb{E}u_i^2 = 1$. Then (i) follows from

$$[\mathbb{E}uu^\top]_{ij} = \mathbb{E}u_i u_j = \begin{cases} \mathbb{E}u_i^2 & \text{if } i = j \\ \mathbb{E}u_i \mathbb{E}u_j & \text{if } i \neq j \end{cases} = \begin{cases} 1 & \text{if } i = j \\ 0 & \text{if } i \neq j \end{cases}.$$

For (ii), we only need to focus on the diagonal since the off-diagonal is already zero.

$$[\mathbb{E}U^2]_{ii} = [\mathbb{E}\mathrm{diag}(u_1, \ldots, u_d)^2]_{ii} = \mathbb{E}u_i^2 = 1.$$

$\square$

**Lemma A.3.** *Suppose Assumption 2.2 holds, $\mathcal{T}_\lambda$ is the reparametrization function for a location-scale family, and the linear parametrization is used. Then*

$$\mathbb{E}\|\mathcal{T}_\lambda(u) - \mathcal{T}_{\lambda'}(u)\|_2^2 = \|\lambda - \lambda'\|_2^2.$$

*Proof.* Denoting $\lambda = (m, C)$ and $\lambda' = (m', C')$,

$$\begin{aligned}
\mathbb{E}&\|\mathcal{T}_\lambda(u) - \mathcal{T}_{\lambda'}(u)\|_2^2 \\
&= \mathbb{E}\|(Cu + m) - (C'u - m')\|_2^2 \\
&= \mathbb{E}\|(C - C')u + (m - m')\|_2^2 \\
&= \mathbb{E}\|(C - C')u\|_2^2 + 2\langle(C - C')\mathbb{E}u, m - m'\rangle + \mathbb{E}\|m - m'\|_2^2 \\
&= \mathbb{E}\|(C - C')u\|_2^2 + \mathbb{E}\|m - m'\|_2^2. \qquad \text{(Assumption 2.2)} \qquad (10)
\end{aligned}$$

Lastly,

$$\begin{aligned}
\mathbb{E}\|(C - C')u\|_2^2 &= \mathbb{E}u^\top(C - C')^\top(C - C')u \\
&= \mathbb{E}\,\mathrm{tr}\,u^\top(C - C')^\top(C - C')u \\
&= \mathrm{tr}\,(C - C')^\top(C - C')\mathbb{E}uu^\top \qquad \text{(cyclic property of trace)} \\
&= \mathrm{tr}\,(C - C')^\top(C - C')I \qquad \text{(Lemma A.2)} \\
&= \mathrm{tr}\,(C - C')^\top(C - C') \\
&= \|C - C'\|_F^2.
\end{aligned}$$

Combining this with Eq. (10) yields the result. $\square$

**Lemma A.4.** *Suppose $f$ is $\mu$-strongly convex and Assumption 2.7 holds. Then $f$ is $L$-Lipschitz smooth, while the constants satisfy the ordering*

$$\mu \quad \leq \quad L \quad \leq \quad \mathcal{L}.$$

*Proof.* For all $\lambda, \lambda' \in \Lambda$, the unbiasedness of $\widehat{\nabla}f$ and Jensen's inequality states that

$$\|\nabla f(\lambda) - \nabla f(\lambda')\|_2^2 = \|\mathbb{E}\widehat{\nabla}f(\lambda; u) - \mathbb{E}\widehat{\nabla}f(\lambda'; u)\|_2^2 \leq \mathbb{E}\|\widehat{\nabla}f(\lambda; u) - \widehat{\nabla}f(\lambda'; u)\|_2^2.$$

Then the $\mu$-strong convexity of $f$ and Assumption 2.7 yields the inequality

$$\mu^2\|\lambda - \lambda'\|_2^2 \leq \|\nabla f(\lambda) - \nabla f(\lambda')\|_2^2 \leq \mathbb{E}\|\widehat{\nabla}f(\lambda; u) - \widehat{\nabla}f(\lambda'; u)\|_2^2 \leq \mathcal{L}^2\|\lambda - \lambda'\|_2^2,$$

from which the statement follows immediately. $\square$

# B Proofs

## B.1 Proofs of Results in Section 2

### B.1.1 Proof of Proposition 2.9

Under the stated assumptions, we first establish a convergence bound which bounds $\mathbb{E}\|\lambda_T - \lambda_*\|_2^2$ after $T$ iterations under a given step size schedule. We will invert this convergence bound into a complexity guarantee by identifying the conditions on $T$, $t_*$, $\tau$, and $\gamma_0$ that guarantee $\mathbb{E}\|\lambda_T - \lambda_*\|_2^2 \leq \epsilon$ for a given $\epsilon > 0$.

**Lemma B.1.** *Suppose $f$ is $\mu$-strongly convex, $h$ satisfies Assumption 2.5, and $\widehat{\nabla}f$ satisfies Assumptions 2.7 and 2.8. Then, for the global optimum $\lambda_* = \arg\min_{\lambda \in \Lambda} F(\lambda)$, and the step size schedule in Eq. (5) with any $t_* \geq 0$ and $\tau \geq 4\mathcal{L}^2/\mu^2$, the contraction coefficient $\rho \triangleq 1 - \mu\gamma_0$ satisfies $\rho \in (0,1)$ and the last iterate of SPGD after $T$ iterations, $\lambda_T$, satisfies*

$$\mathbb{E}\|\lambda_T - \lambda_*\|_2^2 \leq \|\lambda_0 - \lambda_*\|_2^2 \, \rho^{t_*} \frac{(t_* + \tau)^2}{(T + \tau)^2} + 2\gamma_0 \frac{\sigma^2}{\mu} \frac{(t_* + \tau)^2}{(T + \tau)^2} + \frac{8\sigma^2}{\mu^2} \frac{T - t_*}{(T + \tau)^2} \ .$$

*Proof.* The *full proof* is deferred to Appendix B.1.2, p. 22. □

This is a slight generalization of the result by Domke et al. (2023, Theorem 7), where the switching time $t_*$ was fixed to some $t_* \propto \mathcal{L}^2/\mu^2$. While the choice of $t_* \propto \mathcal{L}^2/\mu^2$ results in the typical $\mathrm{O}(1/\epsilon)$ asymptotic complexity, it suffers from a suboptimal polynomial dependence on the initialization error $\Delta = \|\lambda_0 - \lambda_*\|_2$. Picking an alternative $t_*$, which is what we do in the proof, improves the iteration complexity to $\mathrm{O}\big(1/\epsilon + 1/\sqrt{\epsilon}\log\Delta^2 + \log(\Delta^2/\epsilon)\big)$. Now, a similar $\mathrm{O}(1/\epsilon)$ result for vanilla SGD was demonstrated by Stich (2019), where the dependence on $\Delta$ is also logarithmic. Their step size schedule, however, requires the maximum number of iterations $T$, which means $T$ must be fixed before running the algorithm. Our step size, on the other hand, does not require $T$ and is therefore an *any-time* result.

**Proposition 2.9.** *Suppose $f$ is $\mu$-strongly convex, $h$ satisfies Assumption 2.5, and $\widehat{\nabla}f$ satisfies Assumptions 2.7 and 2.8. Then, for the global optimum $\lambda_* = \arg\min_{\lambda \in \Lambda} F(\lambda)$, $\Delta \triangleq \|\lambda_0 - \lambda_*\|_2$, some $t_*$, $\tau$, and $\gamma_0$ (explicit in the proof), SPGD with the step size schedule in Eq. (5) guarantees*

$$T \geq \mathrm{O}\left\{ \frac{\sigma^2}{\mu^2}\frac{1}{\epsilon} + \frac{\sigma\mathcal{L}}{\mu^2}\log\left(\frac{\mathcal{L}^2}{\sigma^2}\Delta^2\right)\frac{1}{\sqrt{\epsilon}} + \frac{\mathcal{L}^2}{\mu^2}\log\left(\Delta^2\frac{1}{\epsilon}\right) + 1 \right\} \quad \Rightarrow \quad \mathbb{E}\|\lambda_T - \lambda_*\|_2^2 \leq \epsilon \ .$$

*Proof.* Since $f$ is strongly convex and $h$ is convex, $F$ is also strongly convex. This implies that, by the property of strictly convex functions, $F$ has a unique global optimum, which we denote as $\lambda_*$. Then we can invoke Lemma B.1. We will optimize the upper bound over the parameters $t_*$, $\tau$, $\gamma_0$, and $T$ so that we can ensure the $\epsilon$-accuracy guarantee $\mathbb{E}\|\lambda_T - \lambda_*\|_2^2 \leq \epsilon$.

Consider the choice

$$t_* = \left\lceil \frac{1}{\log 1/\rho}\log\left(\frac{\mu}{2\gamma_0\sigma^2}\|\lambda_0 - \lambda_*\|_2^2\right) \right\rceil, \qquad \tau = \frac{4\mathcal{L}^2}{\mu^2}, \quad \text{and} \quad \gamma_0 = \frac{\mu}{2\mathcal{L}^2} \ . \tag{11}$$

We will separately analyze the cases of $t_* \geq T$ and $t_* < T$. When $t_* \geq T$, the second-stage never kicks in. Therefore, we can apply Eq. (27) with $t_* = T$. Furthermore,

$$\left\lceil \frac{1}{\log 1/\rho}\log\left(\frac{\mu}{2\gamma_0\sigma^2}\|\lambda_0 - \lambda_*\|_2^2\right) \right\rceil \geq T$$

is true. An immediate implication is that

$$\frac{1}{\log 1/\rho}\log\left(\frac{\mu}{2\gamma_0\sigma^2}\|\lambda_0 - \lambda_*\|_2^2\right) + 1 \geq T$$

$$\Leftrightarrow \qquad \log\left(\frac{\mu}{2\gamma_0\sigma^2}\|\lambda_0 - \lambda_*\|_2^2\right) \geq (\log 1/\rho)(T - 1)$$

$$\Leftrightarrow \qquad \frac{\mu}{2\gamma_0\sigma^2}\|\lambda_0 - \lambda_*\|_2^2 \geq \rho^{-(T-1)}$$

$$\Leftrightarrow \qquad \|\lambda_0 - \lambda_*\|_2^2 \rho^{T-1} \geq 2\gamma_0 \frac{\sigma^2}{\mu} \ . \tag{12}$$

Considering this fact, applying Eq. (27) with $t_* = T$,

$$\mathbb{E}\|\lambda_T - \lambda_*\|_2^2 \leq \|\lambda_0 - \lambda_*\|_2^2 \rho^T + 2\gamma_0 \frac{\sigma^2}{\mu} \qquad (t_* = T)$$

$$\leq \|\lambda_0 - \lambda_*\|_2^2 \rho^T + \|\lambda_0 - \lambda_*\|_2^2 \rho^{T-1} \qquad (\text{Eq. (12)})$$

$$\leq 2\|\lambda_0 - \lambda_*\|_2^2 \rho^{T-1} \ . \qquad (\rho < 1)$$

The number of required steps for achieving the $\epsilon$-accuracy requirement follows from

$$2\|\lambda_0 - \lambda_*\|_2^2 \rho^{T-1} \leq \epsilon$$

$$\Leftrightarrow \qquad 2\|\lambda_0 - \lambda_*\|_2^2 \frac{1}{\epsilon} \leq (1/\rho)^{T-1}$$

$$\Leftrightarrow \qquad \log\left(2\|\lambda_0 - \lambda_*\|_2^2 \frac{1}{\epsilon}\right) \leq (T-1)\log(1/\rho)$$

$$\Leftrightarrow \qquad \frac{1}{\log 1/\rho}\log\left(2\|\lambda_0 - \lambda_*\|_2^2 \frac{1}{\epsilon}\right) \leq T - 1$$

$$\Leftarrow \qquad \frac{1}{1-\rho}\log\left(2\|\lambda_0 - \lambda_*\|_2^2 \frac{1}{\epsilon}\right) \leq T - 1 \qquad (\log(1/\rho) \geq 1 - \rho)$$

$$\Leftrightarrow \qquad \frac{2\mathcal{L}^2}{\mu^2}\log\left(2\|\lambda_0 - \lambda_*\|_2^2 \frac{1}{\epsilon}\right) + 1 \leq T \qquad (1 - \rho = \gamma_0 \mu = \mu^2/(2\mathcal{L}^2)) \tag{13}$$

For the case $t_* < T$,

$$t_* = \left\lceil \frac{1}{\log 1/\rho}\log\left(\frac{\mu}{2\gamma_0\sigma^2}\|\lambda_0 - \lambda_*\|_2^2\right)\right\rceil \tag{14}$$

$$\geq \frac{1}{\log 1/\rho}\log\left(\frac{\mu}{2\gamma_0\sigma^2}\|\lambda_0 - \lambda_*\|_2^2\right)$$

$$= \frac{1}{\log \rho}\log\left(\frac{2\gamma_0\sigma^2}{\mu}\frac{1}{\|\lambda_0 - \lambda_*\|_2^2}\right) \ .$$

This implies

$$\rho^{t_*} \leq \frac{2\gamma_0\sigma^2}{\mu}\frac{1}{\|\lambda_0 - \lambda_*\|_2^2} \ .$$

Substituting for this in Lemma B.1,

$$\mathbb{E}\|\lambda_T - \lambda_*\|_2^2 \leq \|\lambda_0 - \lambda_*\|_2^2 \rho^{t_*}\frac{(t_* + \tau)^2}{T^2} + 2\gamma_0\frac{\sigma^2}{\mu}\frac{(t_* + \tau)^2}{T^2} + \frac{8\sigma^2}{\mu^2}\frac{T - t_*}{T^2} \tag{15}$$

$$\leq 4\gamma_0\frac{\sigma^2}{\mu}\frac{(t_* + \tau)^2}{T^2} + 8\frac{\sigma^2}{\mu^2}\frac{T - t_*}{T^2}$$

$$\leq 4\gamma_0\frac{\sigma^2}{\mu}\frac{(t_* + \tau)^2}{T^2} + 8\frac{\sigma^2}{\mu^2}\frac{1}{T}$$

$$\leq 4\gamma_0\frac{\sigma^2}{\mu}\frac{2t_*^2 + 2\tau^2}{T^2} + 8\frac{\sigma^2}{\mu^2}\frac{1}{T}$$

$$= a\frac{1}{T^2} + b\frac{1}{T} \ ,$$

which is a quadratic function of $1/T$ with the coefficients

$$a \triangleq 8\gamma_0\frac{\sigma^2}{\mu}\left(t_*^2 + \tau^2\right) \qquad \text{and} \qquad b \triangleq 8\frac{\sigma^2}{\mu^2} \ .$$

Achieving the $\epsilon$-accuracy guarantee is equivalent to finding the largest $x = 1/T$ satisfying the inequalities $x > 0$ and

$$ax^2 + bx \leq \epsilon .$$

By the quadratic formula, this is equivalent to finding the largest $x$ satisfying

$$0 \leq x \leq \frac{-b + \sqrt{b^2 + 4a\epsilon}}{2a} .$$

Therefore, picking any

$$T \geq \frac{2a}{-b + \sqrt{b^2 + 4a\epsilon}}$$

is sufficient to obtain an $\epsilon$-accurate solution. To make the bound more interpretable, after defining $\alpha = 4a\epsilon$ and $\beta = b$, we can use the inequality (Symbol-1, 2022)

$$\frac{\alpha}{2\sqrt{\beta^2 + \alpha}} \leq -\beta + \sqrt{\beta^2 + \alpha} .$$

Then

$$\frac{2a}{-b + \sqrt{b^2 + 4a\epsilon}} \quad \leq \quad 2a\frac{2\sqrt{b^2 + 4a\epsilon}}{4a\epsilon} \quad = \quad \sqrt{b^2 + 4a\epsilon}\,\frac{1}{\epsilon} \quad \leq \quad b\frac{1}{\epsilon} + 2\sqrt{a}\frac{1}{\sqrt{\epsilon}} ,$$

where we used the inequality $\sqrt{a + b} \leq \sqrt{a} + \sqrt{b}$. Thus, we have

$$T \geq b\frac{1}{\epsilon} + 2\sqrt{a}\frac{1}{\sqrt{\epsilon}} \qquad \Rightarrow \qquad \mathbb{E}\|\lambda_T - \lambda_*\|_2^2 \leq \epsilon .$$

Substituting $t_*$ and $\gamma_0$ with the expressions in Eq. (11),

$$T \geq 8\frac{\sigma^2}{\mu^2}\frac{1}{\epsilon} + 2\sqrt{8\gamma_0\frac{\sigma^2}{\mu}\left(t_*^2 + \tau^2\right)}\frac{1}{\sqrt{\epsilon}} \tag{16}$$

$$\Leftarrow \quad T \geq 8\frac{\sigma^2}{\mu^2}\frac{1}{\epsilon} + 4\sqrt{2}\sqrt{\gamma_0}\frac{\sigma}{\mu^{1/2}}(t_* + \tau)\frac{1}{\sqrt{\epsilon}}$$

$$\Leftarrow \quad T \geq 8\frac{\sigma^2}{\mu^2}\frac{1}{\epsilon} + 4\sqrt{2}\sqrt{\gamma_0}\frac{\sigma}{\mu^{1/2}}\left(\frac{1}{\log 1/\rho}\log\left(\frac{\mu}{2\gamma_0\sigma^2}\|\lambda_0 - \lambda_*\|_2^2\right) + \frac{4\mathcal{L}^2}{\mu^2} + 1\right)\frac{1}{\sqrt{\epsilon}} \quad \text{(Eq. (14))}$$

$$\Leftarrow \quad T \geq 8\frac{\sigma^2}{\mu^2}\frac{1}{\epsilon} + 4\sqrt{2}\sqrt{\frac{\mu}{2\mathcal{L}^2}}\frac{\sigma}{\mu^{1/2}}\left(\frac{2\mathcal{L}^2}{\mu^2}\log\left(\frac{\mu}{2\sigma^2}\frac{2\mathcal{L}^2}{\mu}\|\lambda_0 - \lambda_*\|_2^2\right) + \frac{4\mathcal{L}^2}{\mu^2} + 1\right)\frac{1}{\sqrt{\epsilon}} \quad (\log 1/\rho \geq 1 - \rho)$$

$$= 8\frac{\sigma^2}{\mu^2}\frac{1}{\epsilon} + \frac{4\sigma}{\mathcal{L}}\left(\frac{2\mathcal{L}^2}{\mu^2}\log\left(\frac{\mathcal{L}^2}{\sigma^2}\|\lambda_0 - \lambda_*\|_2^2\right) + \frac{4\mathcal{L}^2}{\mu^2} + 1\right)\frac{1}{\sqrt{\epsilon}}$$

Considering this, the sufficient condition for $\mathbb{E}\|\lambda_T - \lambda_*\|_2^2 \leq \epsilon$ is now

$$T \geq \frac{8\sigma^2}{\mu^2}\frac{1}{\epsilon} + \left(\frac{8\sigma\mathcal{L}}{\mu^2}\log\left(\frac{\mathcal{L}^2}{\sigma^2}\|\lambda_0 - \lambda_*\|_2^2\right) + \frac{16\sigma\mathcal{L}}{\mu^2} + \frac{4\sigma}{\mathcal{L}}\right)\frac{1}{\sqrt{\epsilon}} . \tag{17}$$

Combining both cases, that is, Eqs. (13) and (17), we have

$$T \geq \max\left\{\frac{8\sigma^2}{\mu^2}\frac{1}{\epsilon} + 4\sigma\left(\frac{2\mathcal{L}}{\mu^2}\log\left(\frac{\mathcal{L}^2}{\sigma^2}\|\lambda_0 - \lambda_*\|_2^2\right) + \frac{4\mathcal{L}}{\mu^2} + \frac{1}{\mathcal{L}}\right)\frac{1}{\sqrt{\epsilon}}, \; \frac{2\mathcal{L}^2}{\mu^2}\log\left(2\|\lambda_0 - \lambda_*\|_2^2\frac{1}{\epsilon}\right) + 1\right\} .$$

This implies the stated result. $\qquad\square$

### B.1.2 Proof of Lemma B.1

The proof closely mirrors the strategy of Garrigos and Gower (2023, Theorem 12.9), which is a combination of previous analyses of SPGD (Gorbunov et al., 2020; Khaled et al., 2023) with the analysis of SGD strongly convex objectives with a decreasing step size schedule (Gower et al., 2019). The main difference is that Garrigos and Gower utilize a different condition on the gradient variance instead of Assumption 2.7. Specifically, they assume that, for all $\lambda, \lambda' \in \Lambda$, there exists some function of $L(u) : \mathrm{supp}(u) \to [0, \infty)$ such that, for each $u \in \mathrm{supp}(u)$, the function $\widehat{\nabla} f(\lambda; u) : \Lambda \to \mathbb{R}^p$ is $L(u)$-smooth with respect to $\lambda$. This then enables the use of the "convex expected smoothness" (Gorbunov et al., 2020; Khaled et al., 2023) condition, which postulates that, for all $\lambda \in \Lambda$, there exists some $\mathcal{L} < \infty$ such that

$$\mathbb{E}\|\widehat{\nabla} f(\lambda; u) - \widehat{\nabla} f(\lambda'; u)\|_2^2 \leq \mathcal{L}^2 \mathrm{D}_f(\lambda, \lambda') , \tag{18}$$

where

$$\mathrm{D}_f(\lambda, \lambda') \triangleq f(\lambda) - f(\lambda') - \langle \nabla f(\lambda'), \lambda - \lambda' \rangle \tag{19}$$

is the Bregman divergence associated with $f$. Note that Assumption 2.7 and the $\mu$-strong convexity of $f$ implies Eq. (18). Therefore, under our assumptions, one can invoke the results that assume Eq. (18), which was the strategy by some previous analyses of BBVI (Kim et al., 2023a; Ko et al., 2024). Here, we will take a more straightforward approach that uses Assumption 2.7 directly in the convergence proof, but the results are identical to the indirect approach of establishing Eq. (18).

**Lemma B.1.** *Suppose $f$ is $\mu$-strongly convex, $h$ satisfies Assumption 2.5, and $\widehat{\nabla} f$ satisfies Assumptions 2.7 and 2.8. Then, for the global optimum $\lambda_* = \arg\min_{\lambda \in \Lambda} F(\lambda)$, and the step size schedule in Eq. (5) with any $t_* \geq 0$ and $\tau \geq 4\mathcal{L}^2/\mu^2$, the contraction coefficient $\rho \triangleq 1 - \mu\gamma_0$ satisfies $\rho \in (0, 1)$ and the last iterate of SPGD after $T$ iterations, $\lambda_T$, satisfies*

$$\mathbb{E}\|\lambda_T - \lambda_*\|_2^2 \leq \|\lambda_0 - \lambda_*\|_2^2 \, \rho^{t_*} \frac{(t_* + \tau)^2}{(T + \tau)^2} + 2\gamma_0 \frac{\sigma^2}{\mu} \frac{(t_* + \tau)^2}{(T + \tau)^2} + \frac{8\sigma^2}{\mu^2} \frac{T - t_*}{(T + \tau)^2} .$$

*Proof.* Since $f$ is strongly convex and $h$ is convex, $F$ is also strongly convex. This implies that $F$ has a unique global optimum, which we denote as $\lambda_*$. Furthermore, under the stated assumptions on $h$, the proximal operator $\mathrm{prox}_{\gamma h}(\cdot)$ is non-expansive for any $\gamma \in (0, \infty)$ (Garrigos and Gower, 2023, Lemma 8.17) and any $\lambda, \lambda' \in \mathbb{R}^p$ such that

$$\|\mathrm{prox}_{\gamma h}(\lambda) - \mathrm{prox}_{\gamma h}(\lambda')\|_2 \leq \|\lambda - \lambda'\|_2 \tag{20}$$

and $\lambda_*$ is the fixed-point of the deterministic proximal gradient descent step (Garrigos and Gower, 2023, Lemma 8.18) such that

$$\mathrm{prox}_{\gamma h}(\lambda_* - \gamma\nabla f(\lambda_*)) = \lambda_* . \tag{21}$$

Using these facts,

$$\|\lambda_{t+1} - \lambda_*\|_2^2 \leq \|\mathrm{prox}_{\gamma_t h}(\lambda_t - \gamma_t \widehat{\nabla} f(\lambda_t; u)) - \mathrm{prox}_{\gamma_t h}(\lambda_* - \gamma_t \nabla f(\lambda_*))\|_2^2 \quad \text{(Eq. (21))}$$

$$\leq \|\lambda_t - \gamma_t \widehat{\nabla} f(\lambda_t; u)) - \lambda_* + \gamma_t \nabla f(\lambda_*)\|_2^2 . \quad \text{(Eq. (20))}$$

Expanding the square,

$$\|\lambda_{t+1} - \lambda_*\|_2^2 \leq \|\lambda_t - \lambda_*\|_2^2 - 2\gamma_t \langle \widehat{\nabla} f(\lambda_t; u) - \nabla f(\lambda_*), \lambda_t - \lambda_* \rangle + \gamma_t^2 \|\widehat{\nabla} f(\lambda_t; u) - \nabla f(\lambda_*)\|_2^2 .$$

Denoting the filtration of the $\sigma$-field of the iterates generated up to iteration $t$ as $\mathcal{F}_t$,

$$\mathbb{E}\big[\|\lambda_{t+1} - \lambda_*\|_2^2 \mid \mathcal{F}_t\big]$$

$$\leq \|\lambda_t - \lambda_*\|_2^2 - 2\gamma_t^2 \Big\langle \mathbb{E}\big[\widehat{\nabla} f(\lambda_t; u) \mid \mathcal{F}_t\big] - \nabla f(\lambda_*), \lambda_t - \lambda_* \Big\rangle + \gamma_t^2 \mathbb{E}\big[\|\widehat{\nabla} f(\lambda_t; u) - \nabla f(\lambda_*)\|_2^2 \mid \mathcal{F}_t\big]$$

$$= \|\lambda_t - \lambda_*\|_2^2 - 2\gamma_t \langle \nabla f(\lambda_t) - \nabla f(\lambda_*), \lambda_t - \lambda_* \rangle + \gamma_t^2 \mathbb{E}\big[\|\widehat{\nabla} f(\lambda_t; u) - \nabla f(\lambda_*)\|_2^2 \mid \mathcal{F}_t\big] , \tag{22}$$

where the equality follows from the fact that $\widehat{\nabla} f$ is unbiased conditional on any $\lambda_t \in \Lambda$.

From the $\mu$-strong convexity of $f$,

$$- 2\gamma_t \langle \nabla f(\lambda_t) - \nabla f(\lambda_*), \lambda_t - \lambda_* \rangle$$

$$= -2\gamma_t \langle \nabla f(\lambda_t), \lambda_t - \lambda_* \rangle + 2\gamma_t \langle \nabla f(\lambda_*), \lambda_t - \lambda_* \rangle$$

$$\leq -\gamma_t \mu \|\lambda_t - \lambda_*\|_2^2 - 2\gamma_t \{ f(\lambda_t) - f(\lambda_*) - \langle \nabla f(\lambda_*), \lambda_t - \lambda_* \rangle \} \quad (\mu\text{-strong convexity of } f)$$

$$= -\gamma_t \mu \|\lambda_t - \lambda_*\|_2^2 - 2\gamma_t \mathrm{D}_f(\lambda_t, \lambda_*) \qquad\qquad\qquad\qquad\qquad (\text{Eq. (19)}) . \qquad (23)$$

The gradient variance at $\lambda_t$, on the other hand, can be compared against the gradient variance at $\lambda_*$ through the variance transfer strategy as

$$\gamma_t^2 \mathbb{E}\Big[ \|\widehat{\nabla f}(\lambda_t; u) - \nabla f(\lambda_*; u)\|_2^2 \mid \mathcal{F}_t \Big]$$

$$= \gamma_t^2 \mathbb{E}\Big[ \|\widehat{\nabla f}(\lambda_t; u) - \widehat{\nabla f}(\lambda_*; u) + \widehat{\nabla f}(\lambda_*; u) - \nabla f(\lambda_*; u)\|_2^2 \mid \mathcal{F}_t \Big]$$

$$\leq 2\gamma_t^2 \mathbb{E}\Big[ \|\widehat{\nabla f}(\lambda_t; u) - \widehat{\nabla f}(\lambda_*; u)\|_2^2 \mid \mathcal{F}_t \Big] + 2\gamma_t^2 \mathbb{E}\Big[ \|\widehat{\nabla f}(\lambda_*; u) - \nabla f(\lambda_*; u)\|_2^2 \mid \mathcal{F}_t \Big] \quad (\text{Young's inequality})$$

$$\leq 2\gamma_t^2 \mathcal{L}^2 \|\lambda_t - \lambda_*\|_2^2 + 2\gamma_t^2 \sigma^2 , \qquad\qquad\qquad\qquad\qquad (\text{Assumptions 2.7 and 2.8})$$

$$= 4\gamma_t^2 \frac{\mathcal{L}^2}{\mu} \Big( f(\lambda_t) - f(\lambda_*) - \langle \nabla f(\lambda_t), \lambda_t - \lambda_* \rangle \Big) + 2\gamma_t^2 \sigma^2 \qquad\qquad (\mu\text{-strong convexity of } f)$$

$$= 4\gamma_t^2 \frac{\mathcal{L}^2}{\mu} \mathrm{D}_f(\lambda_t, \lambda_*) + 2\gamma_t^2 \sigma^2 . \qquad\qquad\qquad\qquad\qquad\qquad (24)$$

Applying Eqs. (23) and (24) to Eq. (22),

$$\mathbb{E}\big[\|\lambda_{t+1} - \lambda_*\|_2^2 \mid \mathcal{F}_t\big] \leq \|\lambda_t - \lambda_*\|_2^2 - \gamma_t\Big(\mu \|\lambda_t - \lambda_*\|_2^2 + 2\,\mathrm{D}_f(\lambda_t, \lambda_*)\Big) + 2\gamma_t^2\Big(2\frac{\mathcal{L}^2}{\mu}\mathrm{D}_f(\lambda_t, \lambda_*) + \sigma^2\Big)$$

$$= (1 - \gamma_t \mu)\|\lambda_t - \lambda_*\|_2^2 + 2\gamma_t\Big(2\gamma_t\frac{\mathcal{L}^2}{\mu} - 1\Big)\mathrm{D}_f(\lambda_t, \lambda_*) + 2\gamma_t^2 \sigma^2 .$$

Taking expectation over all randomness, we obtain our general partial contraction bound

$$\mathbb{E}\|\lambda_{t+1} - \lambda_*\|_2^2 \leq (1 - \gamma_t \mu)\mathbb{E}\|\lambda_t - \lambda_*\|_2^2 + 2\gamma_t\Big(2\gamma_t\frac{\mathcal{L}^2}{\mu} - 1\Big)\mathbb{E}[\mathrm{D}_f(\lambda_t, \lambda_*)] + 2\gamma_t^2 \sigma^2 . \quad (25)$$

Due to the form of the step size schedule, SPGD operates in two different regimes: the first stage with a fixed step size $\gamma_t = \gamma_0$ ($t \in \{0, \ldots, t_* - 1\}$) and the second stage with a decreasing step size $\gamma_{t+1} < \gamma_t$ ($t \in \{t_*, \ldots, T\}$). In the first stage, $\gamma_t = \gamma_0 \leq \frac{\mu}{2\mathcal{L}^2}$. Then the Bregman divergence term in Eq. (25) is negative such that

$$\mathbb{E}\|\lambda_{t+1} - \lambda_*\|_2^2 \leq (1 - \gamma_t \mu)\mathbb{E}\|\lambda_t - \lambda_*\|_2^2 + 2\gamma_t^2 \sigma^2 \qquad\qquad\qquad\qquad (26)$$

Unrolling the recursion yields

$$\mathbb{E}\|\lambda_{t_*} - \lambda_*\|_2^2 \leq (1 - \gamma_0 \mu)^{t_*}\|\lambda_0 - \lambda_*\|_2^2 + 2\gamma_0^2 \sigma^2 \sum_{t=0}^{t_*-1}(1 - \gamma_0 \mu)^t$$

$$\leq (1 - \gamma_0 \mu)^{t_*}\|\lambda_0 - \lambda_*\|_2^2 + 2\gamma_0 \frac{\sigma^2}{\mu} \qquad\qquad (\text{geometric series sum formula})$$

$$\leq \rho^{t_*}\|\lambda_0 - \lambda_*\|_2^2 + 2\gamma_0 \frac{\sigma^2}{\mu} . \qquad\qquad\qquad\qquad\qquad (27)$$

From Lemma A.4, we deduce that $\gamma_0 \mu = \mu^2/(2\mathcal{L}^2) \leq 1/2$, which implies $\rho \in (0, 1)$.

We now turn to the second stage, where the step size starts decreasing. Notice that, for any $\tau \geq 4\mathcal{L}^2/\mu^2$, Eq. (5) satisfies

$$\gamma_t = \frac{1}{\mu}\frac{2(t + \tau) + 1}{(t + \tau + 1)^2} \leq \frac{1}{\mu}\frac{2}{\tau} \leq \frac{1}{\mu}\frac{2\mu^2}{4\mathcal{L}^2} \leq \frac{\mu}{2\mathcal{L}^2} .$$

Therefore, $\gamma_t \leq \frac{\mu}{2\mathcal{L}^2}$ for all $t \geq 0$. Again, the Bregman term in Eq. (25) is negative such that

$$\mathbb{E}\|\lambda_{t+1} - \lambda_*\|_2^2 \leq (1 - \gamma_t \mu)\mathbb{E}\|\lambda_t - \lambda_*\|_2^2 + 2\gamma_t^2 \sigma^2 .$$

Subtituting $\gamma_t$ with the choice in Eq. (5), we obtain

$$\mathbb{E}\|\lambda_{t+1} - \lambda_*\|_2^2 \leq \left(1 - \frac{2(t+\tau)+1}{(t+\tau+1)^2}\right)\mathbb{E}\|\lambda_t - \lambda_*\|_2^2 + 2\frac{\sigma^2}{\mu^2}\frac{(2(t+\tau)+1)^2}{(t+\tau+1)^4}$$

$$= \frac{(t+\tau)^2}{(t+\tau+1)^2}\mathbb{E}\|\lambda_t - \lambda_*\|_2^2 + 2\frac{\sigma^2}{\mu^2}\frac{(2(t+\tau)+1)^2}{(t+\tau+1)^4} \ .$$

Multiplying $(t+\tau+1)^2$ to both sides,

$$(t+\tau+1)^2\mathbb{E}\|\lambda_{t+1} - \lambda_*\|_2^2 \leq (t+\tau)^2\mathbb{E}\|\lambda_t - \lambda_*\|_2^2 + 2\frac{\sigma^2}{\mu^2}\frac{(2(t+\tau)+1)^2}{(t+\tau+1)^2} \ .$$

Let us choose the Lyapunov function $V_t \triangleq (t+\tau+1)^2\mathbb{E}\|\lambda_{t+1} - \lambda_*\|_2^2$. Then the discrete derivative of the Lyapunov,

$$V_{t+1} - V_t \quad \leq \quad 2\frac{\sigma^2}{\mu^2}\frac{(2(t+\tau)+1)^2}{(t+\tau+1)^2} \quad \leq \quad 8\frac{\sigma^2}{\mu^2} \ ,$$

shows that the energy is increasing only by a constant. By integrating the Lyapunov over the time interval $t = t_*, \ldots, T-1$,

$$V_T - V_{t_*} \quad \leq \quad 8\frac{\sigma^2}{\mu^2}(T - t_*)$$

$$\Leftrightarrow \qquad\qquad V_T \quad \leq \quad V_{t_*} + 8\frac{\sigma^2}{\mu^2}(T - t_*)$$

$$\Leftrightarrow \qquad (T+\tau)^2\,\mathbb{E}\|\lambda_T - \lambda_*\|_2^2 \quad \leq \quad (t_*+\tau)^2\mathbb{E}\|\lambda_{t_*} - \lambda_*\|_2^2 + 8\frac{\sigma^2}{\mu^2}(T - t_*)$$

$$\Leftrightarrow \qquad\qquad \mathbb{E}\|\lambda_T - \lambda_*\|_2^2 \quad \leq \quad \frac{(t_*+\tau)^2}{(T+\tau)^2}\mathbb{E}\|\lambda_{t_*} - \lambda_*\|_2^2 + 8\frac{\sigma^2}{\mu^2}\frac{T - t_*}{(T+\tau)^2} \ .$$

Substuting $\|\lambda_{t_*} - \lambda_*\|_2^2$ with the error in Eq. (27),

$$\mathbb{E}\|\lambda_T - \lambda_*\|_2^2 \leq \left(\rho^{t_*}\|\lambda_0 - \lambda_*\|_2^2 + 2\gamma_0\frac{\sigma^2}{\mu}\right)\frac{(t_*+\tau)^2}{(T+\tau)^2} + 8\frac{\sigma^2}{\mu^2}\frac{T - t_*}{(T+\tau)^2}$$

$$= \|\lambda_0 - \lambda_*\|_2^2\,\rho^{t_*}\frac{(t_*+\tau)^2}{(T+\tau)^2} + 2\gamma_0\frac{\sigma^2}{\mu}\frac{(t_*+\tau)^2}{(T+\tau)^2} + \frac{8\sigma^2}{\mu^2}\frac{T - t_*}{(T+\tau)^2} \ , \qquad (28)$$

which is our stated result. $\qquad\qquad\qquad\qquad\qquad\qquad\qquad\qquad\qquad\qquad\qquad\qquad\qquad\qquad\square$

## B.2 Proofs of Results in Section 3

### B.2.1 Proof of Theorem 3.2

**Theorem 3.2.** *Suppose the following hold:*
1. *$\ell$ is $\mu$-strongly convex and satisfies Assumption 3.1 and $\mu \leq \sigma_{\min}(H) \leq \sigma_{\max}(H) \leq L$.*
2. *$h$ satisfies Assumption 2.5.*
3. *$\mathcal{Q}$ is a mean-field location-scale family, where Assumption 2.2 holds.*
4. *$\widehat{\nabla f}$ is the reparametrization gradient.*

*Denote the global optimum $\lambda_* = (m_*, C_*) = \arg\min_{\lambda \in \Lambda} F(\lambda)$, the irreducible gradient noise as $\sigma_*^2 \triangleq \|m_* - \bar{z}\|_2^2 + \|C_*\|_F^2$, and the stationary point of $\ell$ as $\bar{z} \triangleq \arg\min_{z \in \mathbb{R}^d} \ell(z)$. Then, for some $t_*$, $\tau$, and $\gamma_0$ (explicit in the proof), SPGD with the step size schedule in Eq. (5) guarantees*

$$T \geq \mathrm{O}\Big\{g(d, H, \delta, \mu, \varphi)\Big(\sigma_*^2 \epsilon^{-1} + \sigma_* \log\big(\|\lambda_0 - \lambda_*\|_2^2\big)\epsilon^{-1/2}\Big)\Big\} \quad \Rightarrow \quad \mathbb{E}\|\lambda_T - \lambda_*\|_2^2 \leq \epsilon\,,$$

*where*

$$g(d, H, \delta, \mu, \varphi) \triangleq 2\,(1 + r_4)\big(\|H\|_2^2/\mu^2\big) + 4\big(\delta^2/\mu^2\big)\Big((1/2) + r_4 + \mathbb{E}\max_{j=1,\ldots,d} u_j^2\Big)\,.$$

*Proof.* The proof consists of establishing the sufficient conditions of Proposition 2.9 as follows:

    (i) $\ell$ is $\mu$-strongly convex $\quad \Rightarrow \quad$ $f$ is $\mu$-strongly convex.

    (ii) Assumption 3.1 $\quad \Rightarrow \quad$ Assumptions 2.7 and 2.8.

Under the linear parametrization, (i) was established by Domke (2020, Thm. 9). It remains to establish (ii). Therefore, the proof focuses on analyzing the variance of the gradient estimator $\widehat{\nabla f}$.

Since Assumption 3.1 holds, Lemma 4.1 states that, for all $\lambda, \lambda' \in \mathbb{R}^d \times \mathbb{D}^d$, the inequality

$$\mathbb{E}\|\widehat{\nabla f}(\lambda; u) - \widehat{\nabla f}(\lambda'; u)\|_2^2 \leq \Big\{2(1 + r_4)\|H\|_2^2 + 4\delta^2\Big(1/2 + r_4 + \mathbb{E}\max_{j=1,\ldots,d} u_j^2\Big)\Big\}\|\lambda - \lambda'\|_2^2$$

holds. Since $\Lambda \subset \mathbb{R}^d \times \mathbb{D}^d$ under the linear parametrization, this implies we satisfy Assumption 2.7 with

$$\mathcal{L}^2 = 2(1 + r_4)\|H\|_2^2 + 4\delta^2\Big(1/2 + r_4 + \mathbb{E}\max_{j=1,\ldots,d} u_j^2\Big)\,. \tag{29}$$

Furthermore, For the specific choice of $\lambda_* = (m_*, C_*) = \arg\min_{\lambda \in \Lambda} F(\lambda)$ and $\bar{\lambda} = (\bar{z}, 0_{d \times d})$ (which is not part of $\Lambda$), we have the equality

$$\mathbb{E}\|\widehat{\nabla f}(\lambda_*; u) - \widehat{\nabla f}(\bar{\lambda}; u)\|_2^2 = \mathbb{E}\|\widehat{\nabla f}(\lambda_*; u) - \widehat{\nabla f}(\bar{z}; u)\|_2^2 = \mathbb{E}\|\widehat{\nabla f}(\lambda_*; u)\|_2^2\,.$$

This means Lemma 4.1 also implies Assumption 2.8 with the constant

$$\sigma^2 \quad = \quad \mathcal{L}^2\|\lambda_* - \bar{\lambda}\|_2^2 \quad = \quad \mathcal{L}^2\big(\|m_* - \bar{z}\| + \|C_*\|_F^2\big) \quad = \quad \mathcal{L}^2\sigma_*^2\,. \tag{30}$$

We are now able to invoke Proposition 2.9. Substituting $\mathcal{L}$ and $\sigma^2$ in Eq. (28) with the expressions above, we obtain the condition

$$T \geq \frac{\mathcal{L}^2}{\mu^2}\max\Big\{8\sigma_*^2\frac{1}{\epsilon} + 4\sigma_*\Big(2\log\Big(\frac{1}{\sigma_*^2}\|\lambda_0 - \lambda_*\|_2^2\Big) + 4 + \frac{\mu^2}{\mathcal{L}^2}\Big)\frac{1}{\sqrt{\epsilon}}, 2\log\Big(2\|\lambda_0 - \lambda_*\|_2^2\frac{1}{\epsilon}\Big) + 1\Big\}$$

Finally, substituting for Eq. (29) yields our stated result. $\qquad\square$

### B.2.2 Proof of Proposition 3.4

The result follows from a well-known bound on the expected maximum of sub-exponential random variables. We state the proof for completeness.

**Lemma B.2.** *Let $x_1, \ldots, x_d$ be i.i.d. random variables. Suppose there exists some $t > 0$ such that their moment-generating function (MGF) satisfies $M_{x_i}(t) < \infty$. Then*

$$\mathbb{E} \max_{i=1,\ldots,d} x_i \quad \leq \quad \frac{1}{t} \left( \log M_{x_i}(t) + \log d \right) .$$

*Proof.*

$$
\begin{aligned}
\mathbb{E}\left[ t \max_{i=1,\ldots,d} x_i \right] &= \log \exp \left( \mathbb{E}\left[ t \max_{i=1,\ldots,d} x_i \right] \right) \\
&\leq \log \mathbb{E} \exp \left( t \max_{i=1,\ldots,d} x_i \right) && \text{(Jensen's inequality)} \\
&= \log \mathbb{E} \max_{i=1,\ldots,d} \exp(t x_i) \\
&\leq \log \mathbb{E} \sum_{i=1}^{d} \exp(t x_i) \\
&= \log \sum_{i=1}^{d} M_{x_i}(t) && \text{(Definition of MGFs)} \\
&= \log(d M_{x_i}(t)) . && (x_1, \ldots, x_d \text{ are i.i.d.})
\end{aligned}
$$

Dividing both sides by $t$ yields the statement. $\qquad\square$

Applying Lemma B.2 to $u_i^2$ yields the result.

**Proposition 3.4.** *Suppose there exists some $t > 0$ such that the MGF of $u_i^2$ satisfies $M_{u_i^2}(t) < \infty$. Then*

$$\mathbb{E} \max_{i=1,\ldots,d} u_i^2 \quad \leq \quad (1/t) \left( \log M_{u_i^2}(t) + \log d \right) .$$

*For example, if $\varphi$ is a standard Gaussian, then*

$$g(d, H, \delta, \mu, \varphi) \quad \leq \quad 8 \left( \|H\|_2^2 / \mu^2 \right) + \left( \delta^2 / \mu^2 \right) (22 + 16 \log d) .$$

*Proof.* The first part of the statement is a re-statement of Lemma B.2.

For the special case of $u_i \sim \mathcal{N}(0,1)$, we know that $u_i^2 \sim \chi_1^2$ (Johnson et al., 1995, Eq. 29.1), which is the $\chi^2$ distribution with 1 degree of freedom. The MGF of $\chi_1^2$ is given as

$$M_{u_i^2}(t) = (1 - 2t)^{-1/2} \qquad \text{(Johnson et al., 1995, Eq. 29.6)}$$

for $t \in (0, 1/2)$. Then we can invoke Lemma B.2, which suggests

$$\mathbb{E} \max_{i=1,\ldots,d} u_i^2 \leq \min_{t \in (0,1/2)} \frac{1}{t} \left( -\frac{1}{2} \log(1 - 2t) + \log d \right) .$$

Any fixed choice of $t \in (0, 1/2)$ is a valid upper bound. Picking $t = \frac{1}{2}\left(1 - \frac{1}{e}\right) \geq \frac{1}{4}$ yields

$$\mathbb{E} \max_{i=1,\ldots,d} u_i^2 \leq 4 \left( \frac{1}{2} + \log d \right) . \tag{31}$$

Furthermore, the kurtosis of the standard Gaussian is $r_4 = 3$ (Johnson et al., 1994, Eq. 13.11). Plugging $r_4$ and Eq. (31) into $g$ in Theorem 3.2 yields the statement. $\qquad\square$

### B.2.3 Proof of Proposition 3.5

The result follows from the following moment-based bound on the expected maximum of random variables, which is a non-asymptotic refinement of the proof by Rana (2017).

**Lemma B.3.** *Let $x_1, \ldots, x_d$ be i.i.d. non-negative random variables where, for $k \geq 2$, their $k$th moment is finite. That is, $\mathbb{E}x_i^k = r_k < \infty$. Then*

$$\mathbb{E} \max_{i=1,\ldots,d} x_i \quad \leq \quad d^{1/k}(k/(k-1))^{(k-1)/k} r_k^{1/k} \ .$$

*Proof.* For any $\epsilon > 0$, we have

$$
\begin{aligned}
\mathbb{E} \max_{i=1,\ldots,d} x_i &= \int_0^{\epsilon d^{1/k}} \mathbb{P}\left[ \max_{i=1,\ldots,d} x_i \geq t \right] \mathrm{d}t + \int_{\epsilon d^{1/k}}^{\infty} \mathbb{P}\left[ \max_{i=1,\ldots,d} x_i \geq t \right] \mathrm{d}t \\
&\leq \int_0^{\epsilon d^{1/k}} \mathrm{d}t + \int_{\epsilon d^{1/k}}^{\infty} d\, \mathbb{P}\left[ x_i \geq t \right] \mathrm{d}t && \text{(i.i.d. and } \mathbb{P}[\cdot] \leq 1) \\
&= d^{1/k}\left( \epsilon + \frac{1}{k\epsilon^{k-1}} \int_{\epsilon d^{1/k}}^{\infty} k\left(\epsilon d^{1/k}\right)^{k-1} \mathbb{P}\left[ x_i \geq t \right] \mathrm{d}t \right) \\
&\leq d^{1/k}\left( \epsilon + \frac{1}{k\epsilon^{k-1}} \int_{\epsilon d^{1/k}}^{\infty} k t^{k-1} \mathbb{P}\left[ x_i \geq t \right] \mathrm{d}t \right) && (\epsilon d^{1/k} \leq t) \\
&\leq d^{1/k}\left( \epsilon + \frac{1}{k\epsilon^{k-1}} \int_0^{\infty} k t^{k-1} \mathbb{P}\left[ x_i \geq t \right] \mathrm{d}t \right) && \text{(Decreased lower limit of integral)} \ .
\end{aligned}
$$

Now, from the definition of moments, we know that

$$
\begin{aligned}
\int_0^{\infty} k t^{k-1} \mathbb{P}\left[ x_i \geq t \right] \mathrm{d}t &= \int_0^{\infty} \int_{-\infty}^{\infty} k t^{k-1} \mathbb{1}_{x_i > t}\, \mathrm{d}\mathbb{P}[x_i]\, \mathrm{d}t \\
&= \int_{-\infty}^{\infty} \int_0^{\infty} k t^{k-1} \mathbb{1}_{x_i > t}\, \mathrm{d}t\, \mathrm{d}\mathbb{P}[x_i] && \text{(Fubini's Theorem)} \\
&= \int_{-\infty}^{\infty} \int_0^{x_i} k t^{k-1}\, \mathrm{d}t\, \mathrm{d}\mathbb{P}[x_i] \\
&= \int_{-\infty}^{\infty} x_i^k\, \mathrm{d}\mathbb{P}[x_i] \\
&= r_k \ .
\end{aligned}
$$

Therefore,

$$\mathbb{E} \max_{i=1,\ldots,d} x_i \leq d^{1/k}\left( \epsilon + \frac{1}{k\epsilon^{k-1}} r_k \right) \ .$$

The bound is minimized when setting

$$\epsilon = \left( \frac{k-1}{k} r_k \right)^{1/k} \ .$$

Then

$$
\begin{aligned}
\mathbb{E} \max_{i=1,\ldots,d} x_i &\leq d^{1/k}\left( \left( \frac{k-1}{k} r_k \right)^{1/k} + \frac{1}{k} m_k \left( \frac{k-1}{k} r_k \right)^{-(k-1)/k} \right) \\
&= d^{1/k}\left( \left( \frac{k-1}{k} r_k \right)^{1/k} + \frac{1}{k-1} \left( \frac{k-1}{k} r_k \right)^{1/k} \right) \\
&= d^{1/k}\left( 1 + \frac{1}{k-1} \right) \left( \frac{k-1}{k} r_k \right)^{1/k} \\
&= d^{1/k}\left( \frac{k}{k-1} \right)^{(k-1)/k} r_k^{1/k} \ .
\end{aligned}
$$

$\square$

If the $k$th moment of $u_i^2$ is finite, this then immediately implies a polynomial $\mathrm{O}(d^{1/k})$ bound on $g$.

**Proposition 3.5.** *Suppose, for $k \geq 2$, the $k$th moment of $u_i^2$ is finite as $r_{2k} = \mathbb{E}u_i^{2k} < \infty$. Then*
$$\mathbb{E} \max_{i=1,\dots,d} u_i^2 \quad \leq \quad \sqrt{2}\, d^{1/k}\, r_{2k}^{1/k}\, .$$
*For example, if $\varphi$ is a Student-$t$ with $\nu > 4$ degrees of freedom and unit variance, then*
$$g(d, H, \delta, \mu, \varphi) \quad \leq \quad 8(\|H\|_2^2/\mu^2) + (\delta^2/\mu^2)\left(16 + \sqrt{2}\,\nu^3 d^{\frac{2}{\nu-2}}\right)\, .$$

*Proof.* The first part of the statement directly follows from Lemma B.3, where we simplified $\left(k/k-1\right)^{(k-1)/k}$. In particular, for $k \geq 2$, $\left(k/k-1\right)^{(k-1)/k}$ is monotonically decreasing. Since an order $k \geq 2$ moment exists by the assumption on the degrees of freedom, $\left(k/k-1\right)^{(k-1)/k} \leq \sqrt{2}$.

Let's turn to the second part of the statement. We will denote a Student-$t$ distribution with $\nu$-degrees of freedom as $t_\nu$. Since $t_\nu$ does not have unit variance (Johnson et al., 1995, Eq. 28.7a), we have to set the sampling process from $\varphi$ to be
$$u_i \sim \varphi \qquad \Leftrightarrow \qquad u_i \stackrel{\mathrm{d}}{=} \frac{\nu - 2}{\nu} v_i\, , \quad \text{where} \quad v_i \stackrel{\text{i.i.d.}}{\sim} t_\nu\, .$$

Now, it is known that $v_i^2 \stackrel{\mathrm{d}}{=} w_i \sim \mathrm{FDist}(1, \nu_2)$ (Johnson et al., 1995, §28.7), where $\mathrm{FDist}(\nu_1, \nu_2)$ is Fisher's $F$-distribution with $(\nu_1, \nu_2)$ degrees of freedom. The $k$th raw moment of $\mathrm{FDist}(\nu_1, \nu_2)$, denoted as $m_k \triangleq \mathbb{E}w_i^k$, exists up to $2k < \nu_2 = \nu$ and is given as
$$m_k = \left(\frac{\nu_2}{\nu_1}\right)^k \frac{\Gamma(\nu_1/2 + k)}{\Gamma(\nu_1/2)} \frac{\Gamma(\nu_2/2 + k)}{\Gamma(\nu_2/2)}\, . \qquad \text{(Johnson et al., 1995, Eq. 27.43)}$$

This means that we can invoke Lemma B.3 as
$$\mathbb{E} \max_{i=1,\dots,d} u_i^2 \quad = \quad \left(\frac{\nu - 2}{\nu}\right)^2 \mathbb{E} \max_{i=1,\dots,d} w_i \quad \leq \quad \sqrt{2}\left(\frac{\nu - 2}{\nu}\right)^2 d^{1/k} m_k^{1/k}\, ,$$

with any $k < \nu/2$.

For $m_k^{1/k}$, we can use the fact that the gamma function satisfies the recursion $\Gamma(z + 1) = z\Gamma(z)$, which implies $\Gamma(a/2 + k) = \Gamma(a/2) \prod_{i=0}^{k-1}(a/2 + i)$ for any $a > 0$. Therefore,
$$\begin{aligned}
\left(\frac{\Gamma(a/2 + k)}{\Gamma(a/2)}\right)^{1/k} &= \left(\prod_{i=0}^{k-1}\left(\frac{a}{2} + i\right)\right)^{1/k} & \\
&\leq \frac{1}{k}\sum_{i=0}^{k-1}\left(\frac{a}{2} + i\right) & \text{(AM-GM inequality)} \\
&= \frac{a}{2} + \frac{1}{k}\frac{k(k-1)}{2} & \text{(geometric series sum formula)} \\
&= \frac{a + k - 1}{2}\, . &
\end{aligned}$$

Applying this bound to $a = \nu_2 = \nu$ and $a = \nu_1 = 1$ respectively,
$$\begin{aligned}
m_k^{1/k} &= \left(\nu^k \frac{\Gamma(1/2 + k)}{\Gamma(1/2)} \frac{\Gamma(\nu/2 + k)}{\Gamma(\nu/2)}\right)^{1/k} & \\
&\leq \nu \frac{k}{2} \frac{\nu + k - 1}{2} & \\
&< \nu \frac{\nu}{4} \frac{3\nu}{4} & (k < \nu/2) \\
&< \frac{\nu^3}{4}\, . &
\end{aligned}$$

Also, choosing $k = \lceil \nu/2 - 1 \rceil$, we have $d^{1/k} \leq d^{2/(\nu-2)}$. This yields
$$\mathbb{E} \max_{i=1,\dots,d} u_i^2 \quad < \quad \sqrt{2}\left(\frac{\nu - 2}{\nu}\right)^2 d^{\frac{2}{\nu-2}}\frac{\nu^3}{4} \quad < \quad \frac{1}{2\sqrt{2}}\nu^3 d^{\frac{2}{\nu-2}}\, . \tag{32}$$

Lastly, the kurtosis of $u_i = (\nu - 2)/\nu \, v_i$ follows as (Johnson et al., 1995, Eq. 28.5)

$$r_4 \quad = \quad \left(\frac{\nu - 2}{\nu}\right)^4 \mathbb{E}w_i^2 \quad = \quad \left(\frac{\nu - 2}{\nu}\right)^4 \frac{3\nu^2}{(\nu - 2)(\nu - 4)} \quad = \quad 3\frac{(\nu - 2)^3}{\nu^2(\nu - 4)} \quad \leq \quad 3 \,.$$

Plugging the bound in Eq. (32) and the value of $r_4$ into $g$ in Theorem 3.2 yields the statement. $\quad\square$

## B.3 Proofs of Results in Section 4

### B.3.1 Proof of Lemma 4.1

Under the assumption that $\nabla^2 \ell \preceq L \mathrm{I}_d$ and twice differentiability, it is well known that $\nabla^2 \ell \preceq L \mathrm{I}_d \Rightarrow \ell$ is $L$-smooth. We will prove a supporting result analogous to this under Assumption 3.1, which will allow us to bound the relative growth of $\nabla \ell$.

**Lemma B.4.** *Suppose $\ell : \mathbb{R}^d \to \mathbb{R}$ satisfies Assumption 3.1. Then, for any $W \in \mathbb{R}^{d \times d}$ satisfying $\|W\|_2 < \infty$,*

$$\|W(\nabla \ell(z) - \nabla \ell(z'))\|_2 \leq \|WH(z - z')\|_2 + \delta \|W\|_2 \|z - z'\|_2 .$$

*Proof.* The *full proof* is deferred to Appendix B.3.2, p. 32. □

Using this, we can now simplify the $\nabla \ell$ terms in Eq. (6). Applying Lemma B.4 to $V_{\text{loc}}$ with $W = \mathrm{I}_d$ and Young's inequality,

$$
\begin{aligned}
V_{\text{loc}} &\leq \mathbb{E}(\|H(\mathcal{T}_\lambda(u) - \mathcal{T}_{\lambda'}(u))\|_2 + \delta \|\mathcal{T}_\lambda(u) - \mathcal{T}_{\lambda'}(u)\|_2)^2 && \text{(Lemma B.4)} \\
&\leq 2\mathbb{E}\|H(\mathcal{T}_\lambda(u) - \mathcal{T}_{\lambda'}(u))\|_2^2 + 2\delta^2 \mathbb{E}\|\mathcal{T}_\lambda(u) - \mathcal{T}_{\lambda'}(u)\|_2^2 && \text{(Young's inequality)} \\
&\leq 2\|H\|_2^2 \mathbb{E}\|\mathcal{T}_\lambda(u) - \mathcal{T}_{\lambda'}(u)\|_2^2 + 2\delta^2 \mathbb{E}\|\mathcal{T}_\lambda(u) - \mathcal{T}_{\lambda'}(u)\|_2^2 && \text{(Operator norm)} \\
&= 2(\|H\|_2^2 + \delta^2) \mathbb{E}\|\mathcal{T}_\lambda(u) - \mathcal{T}_{\lambda'}(u)\|_2^2 \\
&= 2(\|H\|_2^2 + \delta^2) \|\lambda - \lambda'\|_2^2 . && \text{(Lemma A.3)} \qquad (33)
\end{aligned}
$$

Similarly, applying Lemma B.4 to $V_{\text{scale}}$ with $W = U$ and Young's inequality,

$$
\begin{aligned}
V_{\text{scale}} &\leq \mathbb{E}(\|UH(\mathcal{T}_\lambda(u) - \mathcal{T}_{\lambda'}(u))\|_2 + \delta \|U\|_2 \mathbb{E}\|\mathcal{T}_\lambda(u) - \mathcal{T}_{\lambda'}(u)\|_2)^2 && \text{(Lemma B.4)} \\
&\leq 2 \underbrace{\mathbb{E}\|UH(\mathcal{T}_\lambda(u) - \mathcal{T}_{\lambda'}(u))\|_2^2}_{V_{\text{const}}} + 2\delta^2 \underbrace{\mathbb{E}\|U\|_2^2 \|\mathcal{T}_\lambda(u) - \mathcal{T}_{\lambda'}(u)\|_2^2}_{V_{\text{non-const}}} && \text{(Young's inequality) .}
\end{aligned}
$$

$$(34)$$

$V_{\text{const}}$ corresponds to the constant component of the Hessian $\nabla^2 \ell$, whereas $V_{\text{non-const}}$ corresponds to the non-constant residual. Denote the location and scale parameters of $\lambda$ and $\lambda'$ as

$$\lambda = (m, C) \qquad \text{and} \qquad \lambda' = (m', C') .$$

For $V_{\text{const}}$, we can use the following lemma:

**Lemma B.5.** *Suppose $\mathcal{T}_\lambda$ is the reparameterization operator of a mean-field location-family and Assumption 2.2 holds. Then, for any matrix $H \in \mathbb{R}^{d \times d}$ and any $\lambda, \lambda' \in \mathbb{R}^d \times \mathbb{D}^d$,*

$$\|UH(\mathcal{T}_\lambda(u) - \mathcal{T}_{\lambda'}(u))\|_2^2 \leq r_4 \|H\|_2^2 \|\lambda - \lambda'\|_2^2 .$$

See the *full proof* in Appendix B.3.3, p. 33.

The remaining part of the proof closely resembles the proof sketch of Lemma 4.1. For convenience, we first restate Lemma 4.1 and then proceed to the full proof.

**Lemma 4.1.** *Suppose Assumptions 2.2 and 3.1 hold, $\mathcal{Q}$ is a mean-field location-family, and $\widehat{\nabla} f$ is the reparametrization gradient. Then, for any $\lambda, \lambda' \in \mathbb{R}^d \times \mathbb{D}^d$.*

$$\mathbb{E}\|\widehat{\nabla} f(\lambda; u) - \widehat{\nabla} f(\lambda'; u)\|_2^2 \leq \left\{ 2(1 + r_4)\|H\|_2^2 + 4\delta^2 \left( 1/2 + r_4 + \mathbb{E} \max_{j=1,\ldots,d} u_j^2 \right) \right\} \|\lambda - \lambda'\|_2^2 .$$

*Proof.* Recall Eq. (34). The proof consists of bounding the two terms $V_{\text{const}}$ and $V_{\text{non-const}}$. First, for $V_{\text{const}}$, under Assumption 3.1,

$$V_{\text{const}} \leq r_4 \|H\|_2^2 \|\lambda - \lambda'\|_2^2 . \qquad \text{(Lemma B.5)} \qquad (35)$$

It remains to bound $V_{\text{non-const}}$, which is our main challenge.

Denote $\bar{m} \triangleq m - m'$ and $\bar{C} \triangleq C - C'$ such that

$$
\begin{aligned}
\mathcal{T}_\lambda(u) - \mathcal{T}_{\lambda'}(u) &= (Cu + m) - (C'u + m') \\
&= (C - C')u + (m - m')
\end{aligned}
$$

$$= \bar{C}\boldsymbol{u} + \bar{m} \ .$$

Then

$$
\begin{aligned}
V_{\text{non-const}} &= \mathbb{E}\|\boldsymbol{U}\|_2^2 \|\mathcal{T}_\lambda(\boldsymbol{u}) - \mathcal{T}_{\lambda'}(\boldsymbol{u})\|_2^2 \\
&= \mathbb{E}\|\boldsymbol{U}\|_2^2 \|\bar{C}\boldsymbol{u} + \bar{m}\|_2^2 \\
&\leq \mathbb{E}\|\boldsymbol{U}\|_2^2 \big(2\|\bar{C}\boldsymbol{u}\|_2^2 + 2\|\bar{m}\|_2^2\big) && \text{(Young's inequality)} \\
&= \mathbb{E}\Big(\max_{j=1,\dots,d} \boldsymbol{u}_j^2\Big) \sum_{i=1}^d \big(2\bar{C}_{ii}^2 \boldsymbol{u}_i^2 + 2\bar{m}_i^2\big) \\
&= 2\mathbb{E}\sum_{i=1}^d \Big(\max_{j=1,\dots,d} \boldsymbol{u}_j^2\Big) \bar{C}_{ii}^2 \boldsymbol{u}_i^2 + 2\mathbb{E}\Big(\max_{j=1,\dots,d} \boldsymbol{u}_j^2\Big)\sum_{i=1}^d \bar{m}_i^2 \ .
\end{aligned}
$$

We will focus on the first term. Denoting $i_* = \arg\max_{i=1,\dots,d} \boldsymbol{u}_i^2$, the coordinate of maximum magnitude, we can decompose the expectation by the contribution of the event $i_* = i$ and $i_* \neq i$. That is,

$$
\mathbb{E}\boldsymbol{u}_{i_*}^2 \sum_{i=1}^d \bar{C}_{ii}^2 \boldsymbol{u}_i^2 = \sum_{i=1}^d \bar{C}_{ii}\, \mathbb{E}\Big[\underbrace{\boldsymbol{u}_{i_*}^4 \mathbb{1}\{i_* = i\}}_{V_{\text{max}}} + \underbrace{\boldsymbol{u}_{i_*}^2 \boldsymbol{u}_i^2 \mathbb{1}\{i_* \neq i\}}_{V_{\text{non-max}}}\Big] \ .
$$

The expectation of the event $i_* = i$ follows as

$$
\begin{aligned}
V_{\text{max}} &= \mathbb{E}\big[\boldsymbol{u}_{i_*}^4 \mathbb{1}\{i_* = i\}\big] \\
&= \mathbb{E}\big[\boldsymbol{u}_{i_*}^4\big]\mathbb{E}[\mathbb{1}\{i_* = i\}] && (\boldsymbol{u}_{i_*} \perp\!\!\!\perp i_*) \\
&= \mathbb{E}\big[\boldsymbol{u}_{i_*}^4\big]\mathbb{P}[i_* = i] \\
&= \mathbb{E}\big[\boldsymbol{u}_{i_*}^4\big]\frac{1}{d} \\
&\leq \mathbb{E}\Big[\sum_{i=1}^d \boldsymbol{u}_i^4\Big]\frac{1}{d} && (\max_{j=1,\dots,d} \boldsymbol{u}_j^4 \leq \sum_{j=1}^d \boldsymbol{u}_j^4) \\
&= (dr_4)\frac{1}{d} && \text{(Assumption 2.2)} \\
&= r_4 \ . && (36)
\end{aligned}
$$

On the other hand, for the event $i_* \neq i$,

$$
\begin{aligned}
V_{\text{non-max}} &= \mathbb{E}\big[\boldsymbol{u}_{i_*}^2 \boldsymbol{u}_i^2 \mathbb{1}\{i_* \neq i\}\big] \\
&= \mathbb{E}\Big[\max_{j\neq i} \boldsymbol{u}_j^2 \boldsymbol{u}_i^2 \mathbb{1}\{i_* \neq i\}\Big] \\
&= \mathbb{E}\Big[\max_{j\neq i} \boldsymbol{u}_j^2 \boldsymbol{u}_i^2\Big] && (\mathbb{1} \leq 1) \\
&\leq \mathbb{E}\Big[\max_{j\neq i} \boldsymbol{u}_j^2\Big]\mathbb{E}\big[\boldsymbol{u}_i^2\big] && (\boldsymbol{u}_j \perp\!\!\!\perp \boldsymbol{u}_i \text{ for all } i \neq j) \\
&= \mathbb{E}\Big[\max_{j=1,\dots,d-1} \boldsymbol{u}_j^2\Big]\mathbb{E}\big[\boldsymbol{u}_i^2\big] && (\boldsymbol{u}_1, \dots, \boldsymbol{u}_d \text{ are i.i.d.}) \\
&= \mathbb{E}\max_{j=1,\dots,d-1} \boldsymbol{u}_j^2 \ . && \text{(Assumption 2.2)}
\end{aligned}
$$

Therefore, we finally obtain

$$
\begin{aligned}
V_{\text{non-const}} &\leq 2\sum_{i=1}^d \Big[\Big(\mathbb{E}\max_{j=1,\dots,d-1} \boldsymbol{u}_j^2 + r_4\Big)\bar{C}_{ii}^2 + \mathbb{E}\max_{j=1,\dots,d} \boldsymbol{u}_j^2 \bar{m}_i^2\Big] \\
&\leq 2\Big(\mathbb{E}\max_{j=1,\dots,d} \boldsymbol{u}_j^2 + r_4\Big)\big(\|\bar{m}\|_2^2 + \|\bar{C}\|_F^2\big) && (\max_{j=1,\dots,d-1} \boldsymbol{u}_j^2 \leq \max_{j=1,\dots,d} \boldsymbol{u}_j^2) \\
&= 2\Big(\mathbb{E}\max_{j=1,\dots,d} \boldsymbol{u}_j^2 + r_4\Big)\|\lambda - \lambda'\|_2^2 \ . && (37)
\end{aligned}
$$

Combining Eqs. (6), (33) to (35) and (37) yields the statement. $\qquad\square$

### B.3.2 Proof of Lemma B.4

**Lemma B.4.** *Suppose $\ell : \mathbb{R}^d \to \mathbb{R}$ satisfies Assumption 3.1. Then, for any $W \in \mathbb{R}^{d \times d}$ satisfying $\|W\|_2 < \infty$,*

$$\|W(\nabla\ell(z) - \nabla\ell(z'))\|_2 \le \|WH(z - z')\|_2 + \delta\|W\|_2\|z - z'\|_2 .$$

*Proof.* From twice differentiability of $\ell$ (Assumption 3.1) and the fundamental theorem of calculus, we know that

$$\|W(\nabla\ell(z) - \nabla\ell(z'))\|_2 = \left\|W \int_0^1 \nabla^2\ell(tz + (1 - t)z')(z - z') \, \mathrm{d}t\right\|_2 .$$

Denoting $z_t \triangleq tz + (1 - t)z'$ for clarity,

$$\|W(\nabla\ell(z) - \nabla\ell(z'))\|_2$$

$$= \left\|\int_0^1 W\nabla^2\ell(z_t)(z - z') \, \mathrm{d}t\right\|_2$$

$$\le \int_0^1 \|W\nabla^2\ell(z_t)(z - z')\|_2 \, \mathrm{d}t \qquad\qquad \text{(Jensen's inequality)}$$

$$= \int_0^1 \left\|W\left(\nabla^2\ell(z_t) - H + H\right)(z - z')\right\|_2 \, \mathrm{d}t$$

$$\le \int_0^1 \left\{\|WH(z - z')\|_2 + \|W\|_2\left\|\nabla^2\ell(z_t) - H\right\|_2\|z - z'\|_2\right\} \, \mathrm{d}t \qquad \text{(Triangle inequality)}$$

$$\le \int_0^1 \left\{\|WH(z - z')\|_2 + \delta\|W\|_2\|z - z'\|_2\right\} \, \mathrm{d}t \qquad\qquad \text{(Assumption 3.1)}$$

$$= \|WH(z - z')\|_2 + \delta\|W\|_2\|z - z'\|_2 .$$

$\square$

### B.3.3 Proof of Lemma B.5

**Lemma B.5.** *Suppose $\mathcal{T}_\lambda$ is the reparameterization operator of a mean-field location-family and Assumption 2.2 holds. Then, for any matrix $H \in \mathbb{R}^{d \times d}$ and any $\lambda, \lambda' \in \mathbb{R}^d \times \mathbb{D}^d$,*

$$\|UH(\mathcal{T}_\lambda(u) - \mathcal{T}_{\lambda'}(u))\|_2^2 \leq r_4 \|H\|_2^2 \|\lambda - \lambda'\|_2^2 .$$

*Proof.* For clarity, let us denote $\bar{C} \triangleq C - C'$ and $\bar{m} \triangleq m - m'$ such that

$$
\begin{aligned}
\mathcal{T}_\lambda(u) - \mathcal{T}_{\lambda'}(u) &= (Cu + m) - (C'u + m') \\
&= (C - C')u + (m - m') \\
&= \bar{C}u + \bar{m} .
\end{aligned}
$$

Then

$$
\begin{aligned}
\|UH(\mathcal{T}_\lambda(u) - \mathcal{T}_{\lambda'}(u))\|_2^2 &= \|UH(\bar{C}u + \bar{m})\|_2^2 \\
&= \underbrace{\mathbb{E}\|UH\bar{C}u\|_2^2}_{V_{\text{scale}}} + 2\underbrace{\langle UH\bar{m}, H\bar{C}u \rangle}_{V_{\text{cross}}} + \underbrace{\mathbb{E}\|UH\bar{m}\|_2^2}_{V_{\text{loc}}} .
\end{aligned}
$$

$V_{\text{loc}}$ and $V_{\text{cross}}$ are straightforward. Under Assumption 2.2, it immediately follows that

$$
\begin{aligned}
V_{\text{loc}} &= \mathbb{E}\|UH\bar{m}\|_2^2 \\
&= \bar{m}^\top H^\top \mathbb{E}U^2 H\bar{m} \\
&= \bar{m}^\top H^\top H\bar{m} \qquad \text{(Lemma A.2)} \\
&= \|H\bar{z}\|_2^2 .
\end{aligned}
$$

On the other hand,

$$
\begin{aligned}
V_{\text{cross}} &= \mathbb{E}\langle UH\bar{m}, UH\bar{C}u \rangle \\
&= \bar{m}^\top H^\top (\mathbb{E}U^2 H\bar{C}u) .
\end{aligned}
$$

The expectation follows as

$$
\begin{aligned}
\left[\mathbb{E}U^2 H\bar{C}u\right]_i &= \mathbb{E}u_i^2 \sum_{j=1}^d H_{ij}\bar{C}_{jj}u_j \\
&= H_{ii}\bar{C}_{ii}\mathbb{E}u_i^3 + \sum_{j \neq i} H_{ij}\bar{C}_{jj}\mathbb{E}u_i^2 \mathbb{E}u_j \\
&= 0 . \qquad \text{(Assumption 2.2)}
\end{aligned}
$$

Thus, the cross term $V_{\text{cross}}$ vanishes.

$V_{\text{scale}}$ requires careful elementwise inspection in order to apply Assumption 2.2. That is,

$V_{\text{scale}} = \mathbb{E}\|UH\bar{C}u\|_2^2$

$$
\begin{aligned}
&= \mathbb{E}\sum_{i=1}^d u_i^2 \left\{\sum_{j=1}^d H_{ij}\bar{C}_{jj}u_j\right\}^2 \\
&= \mathbb{E}\sum_{i=1}^d u_i^2 \left\{H_{ii}\bar{C}_{ii}u_i + \sum_{j \neq i} H_{ij}\bar{C}_{jj}u_j\right\}^2 \\
&= \mathbb{E}\sum_{i=1}^d u_i^2 \left\{H_{ii}^2\bar{C}_{ii}^2 u_i^2 + 2H_{ii}\bar{C}_{ii}u_i\left(\sum_{j \neq i} H_{ij}\bar{C}_{jj}u_j\right) + \left(\sum_{j \neq i} H_{ij}\bar{C}_{jj}u_j\right)^2\right\} && \text{(expand quadratic)} \\
&= \sum_{i=1}^d \left\{H_{ii}^2\bar{C}_{ii}^2\mathbb{E}u_i^4 + 2H_{ii}\bar{C}_{ii}\mathbb{E}u_i^3\mathbb{E}\left(\sum_{j \neq i} H_{ij}\bar{C}_{jj}u_j\right) + \mathbb{E}u_i^2\mathbb{E}\left(\sum_{j \neq i} H_{ij}\bar{C}_{jj}u_j\right)^2\right\} && \text{(distribute } u_i^2\text{)}
\end{aligned}
$$

$$
= \sum_{i=1}^{d} \left\{ r_4 H_{ii}^2 \bar{C}_{ii}^2 + \mathbb{E} \left( \sum_{j \neq i} H_{ij} \bar{C}_{jj} u_j \right)^2 \right\} \qquad \text{(Assumption 2.2)}
$$

$$
= \sum_{i=1}^{d} \left\{ r_4 H_{ii}^2 \bar{C}_{ii}^2 + \sum_{j \neq i} \left( H_{ij}^2 \bar{C}_{jj}^2 \mathbb{E} u_j^2 + \sum_{k \neq j} H_{ij} \bar{C}_{jj} \mathbb{E} u_j H_{ik} \bar{C}_{kk} \mathbb{E} u_k \right) \right\} \qquad \text{(expand quadratic)}
$$

$$
= \sum_{i=1}^{d} \left\{ r_4 H_{ii}^2 \bar{C}_{ii}^2 + \sum_{j \neq i} H_{ij}^2 \bar{C}_{jj}^2 \right\} \qquad \text{(Assumption 2.2)}
$$

$$
= \sum_{i=1}^{d} \sum_{j=1}^{d} H_{ij}^2 \bar{C}_{jj}^2 + (r_4 - 1) \sum_{i=1}^{d} H_{ii}^2 \bar{C}_{ii}^2
$$

$$
= \| H\bar{C} \|_{\mathrm{F}}^2 + (r_4 - 1) \| \mathrm{diag}(H\bar{C}) \|_{\mathrm{F}}^2 \,.
$$

Combining everything,

$$
\| H(\mathcal{T}_\lambda(u) - z) \|_{U^2}^2 = V_{\mathrm{loc}} + 2 V_{\mathrm{cross}} + V_{\mathrm{scale}}
$$
$$
= \| H\bar{m} \|_2^2 + \| H\bar{C} \|_{\mathrm{F}}^2 + (r_4 - 1) \| \mathrm{diag}(H\bar{C}) \|_{\mathrm{F}}^2 \qquad (38)
$$

From the property of the Frobenius norm, for any matrix $A \in \mathbb{R}^{d \times d}$, we can decompose

$$
\| A \|_{\mathrm{F}}^2 \quad = \quad \sum_{i=1}^{d} \sum_{j=1}^{d} A_{ij}^2 \quad = \quad \sum_{i=1}^{d} A_{ii}^2 + \sum_{i=1}^{d} \sum_{i \neq j} A_{ij}^2 \quad = \quad \| \mathrm{diag}(A) \|_{\mathrm{F}}^2 + \| \mathrm{off}(A) \|_{\mathrm{F}}^2 \,,
$$

where $\mathrm{off}(A)$ is a function that zeroes-out the diagonal of $A$. Then from Eq. (38),

$$
\begin{aligned}
\| H(\mathcal{T}_\lambda(u) - z) \|_{U^2}^2 &= \| H\bar{m} \|_2^2 + \| \mathrm{off}(H\bar{C}) \|_{\mathrm{F}}^2 + \| \mathrm{diag}(H\bar{C}) \|_{\mathrm{F}}^2 + (r_4 - 1) \| \mathrm{diag}(H\bar{C}) \|_{\mathrm{F}}^2 \\
&= \| H\bar{m} \|_2^2 + \| \mathrm{off}(H\bar{C}) \|_{\mathrm{F}}^2 + r_4 \| \mathrm{diag}(H\bar{C}) \|_{\mathrm{F}}^2 \\
&\leq r_4 \| H\bar{m} \|_2^2 + r_4 \| \mathrm{off}(H\bar{C}) \|_{\mathrm{F}}^2 + r_4 \| \mathrm{diag}(H\bar{C}) \|_{\mathrm{F}}^2 \qquad \text{(Lemma A.1)} \\
&= r_4 \left( \| H\bar{m} \|_2^2 + \| H\bar{C} \|_{\mathrm{F}}^2 \right) \\
&\leq r_4 \| H \|_2^2 \left( \| \bar{m} \|_2^2 + \| \bar{C} \|_{\mathrm{F}}^2 \right) \qquad \text{(operator norm)} \\
&= r_4 \| H \|_2^2 \| \lambda - \lambda' \|_2^2 \,,
\end{aligned}
$$

which is the stated result. $\qquad \square$

### B.3.4 Proof of Proposition 4.2

For any $\mu, L \in (0, \infty)$ such that $\mu \leq L$, our goal is to obtain a matrix-valued function $H_{\mathrm{worst}} :$ $\mathbb{R}^d \to \mathbb{S}^d_{\succ 0}$ satisfying

$$\mu \mathrm{I}_d \quad \preceq \quad H_{\mathrm{worst}} \quad \preceq \quad L \mathrm{I}_d$$

that, under the choice $H = H_{\mathrm{worst}}$, maximizes the quantity

$$\left\| U \int_0^1 H(z^w)(z - \bar{z}) \mathrm{d}w \right\|_2^2 , \tag{39}$$

where $\bar{z} \in \{z \mid \nabla \ell(z) = 0\}$ is any stationary point of $\ell$, $z \triangleq \mathcal{T}_\lambda(u)$, and $z^w \triangleq wz + (1-w)\bar{z}$. Given the norm constraint, the worst-case example that maximizes Eq. (39) will be the matrix-valued function that approximately results in

$$\left\| U \int_0^1 H(z^w)(z - \bar{z}) \mathrm{d}w \right\|_2^2 \quad \asymp \quad L^2 \|U\|_2^2 \|z - \bar{z}\|_2^2$$

for *any* realization of $u$ on $\mathbb{R}^d$. For this, we will establish the relations

$$\left\| U \int_0^1 H(z^w)(z - \bar{z}) \mathrm{d}w \right\|_2^2 \quad = \quad \|U H(z^w)(z - \bar{z})\|_2^2 \quad \asymp \quad L \|U\|_2^2 \|z - \bar{z}\|_2^2 . \tag{40}$$

The first equality in Eq. (40) follows from identifying the conditions where $H(z^w)$ is independent of the value of $w$. For the specific choice of

$$m = \bar{z} = 0_d, \qquad C = \mathrm{diag}(\delta, \ldots, \delta), \qquad \text{any } \delta > 0 ,$$

$H(z^w)$ is independent of $w$ if it only depends on the quantities

$$i_* = \arg\max_{i=1,\ldots,d} |z_i^w| \qquad \text{and} \qquad \hat{z}^w \triangleq \frac{z^w}{\|z^w\|_2} . \tag{41}$$

That is, with some abuse of notation, $H(z^w) = H(\hat{z}^w, i_*)$.

**Lemma B.6.** *Suppose $m = \bar{z} = 0_d$, and for any $\delta > 0$, $C = \mathrm{diag}(\delta, \ldots, \delta)$. If $H(z^w)$ is a function of only $i_*$ and $\hat{z}^w$, then $H(z^w)$ is constant with respect to $w \in [0, 1]$.*

*Proof.* It suffices to show that, under the stated conditions, the values of $i_*$ and $\hat{z}^w$ are invariant to $w$. For $\hat{z}^w$, this trivially follows from the assumption that $\bar{z} = 0$ as

$$\hat{z}^w = \frac{z^w}{\|z^w\|_2} = \frac{wz + (1-w)\bar{z}}{\|wz + (1-w)\bar{z}\|_2} = \frac{wz}{\|wz\|_2} = \frac{z}{\|z\|_2} .$$

For $i_*$, we use the fact that the diagonal matrix $C$ is isotropic as

$$\arg\max_{i=1,\ldots,d} |z_i^w| \quad = \quad \arg\max_{i=1,\ldots,d} w C_{ii} |u_i| \quad = \quad \arg\max_{i=1,\ldots,d} w\delta |u_i| \quad = \quad \arg\max_{i=1,\ldots,d} |u_i| .$$

$\square$

From $H(z^w) = H(\hat{z}^w, i_*)$, the integral in Eq. (39) can be solved as

$$\left\| U \int_0^1 H(z^w)(z - \bar{z}) \mathrm{d}w \right\|_2^2 \quad = \quad \|U H(z^w)(z - \bar{z})\|_2^2 .$$

It remains to construct $H$ in a way that depends only on $\hat{z}^w$ and $i_*$ such that

$$\|U H(z^w)(z - \bar{z})\|_2^2 \quad \asymp \quad L \|U\|_2^2 \|z - \bar{z}\|_2^2 .$$

Recalling the spectral constraints, this is equivalent to, for all $z \in \mathbb{R}^d$, $H$ solving the equation

$$H(i_*)z = L \|z\|_2 \, e_{i_*} \quad \text{subject to} \quad \mu \mathrm{I}_d \leq H(z^w) \leq L \mathrm{I}_d . \tag{42}$$

Notice the equivalence

$$H(z^w)z = L\,\|z\|_2 e_{i_*} \qquad \Leftrightarrow \qquad H(z^w)\frac{z^w}{\|z^w\|_2} = Le_{i_*}\ .$$

Thus, $\hat{z}^w$ and $i_*$ contain all the information we need. The following matrix-valued function almost solves Eq. (42):

$$H_{\text{worst}}(z) = \alpha I_d + \frac{\beta}{2}\big(e_{i_*}\hat{z}^\top + \hat{z}\,e_{i_*}^\top\big), \quad \text{where} \quad \hat{z} = \frac{z}{\|z\|_2}\ . \tag{43}$$

This function is reminiscent of a Householder reflector ([Trefethen and Bau](), 1997, Eq. 10.4), where some modifications were made to make it satisfy the eigenvalue constraint. From the fact that both $e_{i_*}$ and $\hat{z}$ have a unit norm, it is apparent that this matrix satisfies Assumption 3.1 with $H = \alpha I_d$ and $\delta = \beta$. Furthermore, by setting the constants as

$$\alpha = \frac{L+\mu}{2} \quad \text{and} \quad \beta = \frac{L-\mu}{2}\ , \tag{44}$$

the triangle inequality asserts that the eigenvalue constraint $\mu I_d \preceq H_{\text{worst}} \preceq L I_d$ is satisfied almost surely.

Given the specific form of $H_{\text{worst}}$, we are now ready to formally prove Proposition 4.2. Let us first restate the proposition for convenience and then proceed to the proof.

**Proposition 4.2.** *Suppose Assumption 2.2 holds and $\mathcal{Q}$ is a mean-field location-scale family. Then, for any $t > 0$, $d > 0$, $\mu, L \in (0, +\infty)$ satisfying $\mu \leq L$, there exists a matrix-valued function $H(z) : \mathbb{R}^d \to \mathbb{S}^d_{\succ 0}$ satisfying $\mu I_d \preceq H \preceq L I_d$ almost surely and a set of parameters $\lambda = (m, C) \in \mathbb{R}^d \times \mathbb{D}^d_{\succ 0}$ such that*

$$\mathbb{E}\Big\|U \int_0^1 H(z^w)(z - \bar{z})\mathrm{d}w\Big\|_2^2 \geq \left\{\frac{(L-\mu)^2}{4} - \frac{L^2}{2}\frac{\mathbb{E}\max_{i=1,\dots,d} u_i^4}{d}\right\}c(t,\varphi)\Big\{\mathbb{E}\max_{i=1,\dots,d-1} u_i^2 - t\Big\}\|C\|_F^2\ .$$

*where $c(t, \varphi) > 0$ is a constant only dependent on $t$ and $\varphi$.*

*Proof.* Recall $H_{\text{worst}}$ in Eq. (43). By inspection, we know that $H_{\text{worst}}(z^w)$ only depends on the quantities $i_*$ and $z^w$. Then Lemma B.6 states that $w \mapsto H_{\text{worst}}(z^w)$ is a constant function. Therefore,

$$\mathbb{E}\Big\|U \int_0^1 H_{\text{worst}}(z^w)(z-\bar{z})\mathrm{d}w\Big\|_2^2 = \mathbb{E}\big\|UH_{\text{worst}}\big(z^0\big)(z-\bar{z})\big\|_2^2 \qquad \text{(Lemma B.6)}$$

$$= \mathbb{E}\Big\|U\Big(\alpha I_d + \frac{\beta}{2}\big(e_{i_*}\hat{z}^\top + \hat{z}\,e_{i_*}^\top\big)\Big)z\Big\|_2^2\ . \qquad \text{(Eq. (43))} \tag{45}$$

This can be decomposed as

$$\mathbb{E}\Big\|U\Big(\alpha I_d + \frac{\beta}{2}\big(e_{i_*}\hat{z}^\top + \hat{z}\,e_{i_*}^\top\big)\Big)z\Big\|_2^2$$

$$= \mathbb{E}\Big\|\alpha Uz + \frac{\beta}{2}Ue_{i_*}\big(\hat{z}^\top z\big) + \frac{\beta}{2}U\hat{z}\big(e_{i_*}^\top z\big)\Big\|_2^2$$

$$= \mathbb{E}\Big\|\alpha Uz + \frac{\beta}{2}Ue_{i_*}\|z\|_2 + \frac{\beta}{2}U\hat{z}z_{i_*}\Big\|_2^2$$

$$= \mathbb{E}\Big\|\alpha Uz + \Big(\frac{\beta}{2}\|z\|_2\Big)Ue_{i_*} + \Big(\frac{\beta}{2}\hat{z}_{i_*}\Big)Uz\Big\|_2^2$$

$$= \mathbb{E}\Big\|\Big(\frac{\beta}{2}\|z\|_2\Big)Ue_{i_*} + \Big(\alpha + \frac{\beta}{2}\hat{z}_{i_*}\Big)Uz\Big\|_2^2$$

$$= \mathbb{E}\left[\frac{\beta^2}{4}\|z\|_2^2\|Ue_{i_*}\|_2^2 + \Big(\alpha + \frac{\beta}{2}\hat{z}_{i_*}\Big)^2\|Uz\|_2^2 + \beta\Big(\alpha + \frac{\beta}{2}\hat{z}_{i_*}\Big)\|z\|_2\big(e_{i_*}^\top U^2 z\big)\right]$$

$$= \mathbb{E}\left[\frac{\beta^2}{4}u_{i_*}^2\|z\|_2^2\right] + \underbrace{\mathbb{E}\left[\left(\alpha + \frac{\beta}{2}\hat{z}_{i_*}\right)^2\|Uz\|_2^2\right]}_{\triangleq V_1} + \underbrace{\mathbb{E}\left[\beta\left(\alpha + \frac{\beta}{2}\hat{z}_{i_*}\right)\|z\|_2(\mathrm{e}_{i_*}^\top U^2 z)\right]}_{\triangleq V_2}. \quad (46)$$

Here, the first term $\beta^2/4 u_{i_*}^2\|z\|_2^2$ is the worst-case behavior we expect from solving Eq. (42). The remaining terms $V_1$ and $V_2$ are the error caused by inexactly solving Eq. (42). It suffices to show that $\beta^2/4 u_{i_*}^2\|z\|_2^2$ dominates lower bounds on $V_1$ and $V_2$ asymptotically in $L$ and $d$.

$V_1 \geq 0$ trivially holds and can immediately be lower-bounded. $V_2$, on the other hand, is not necessarily non-negative. Therefore, we will use the bound $V_2 \geq -|\mathbb{E}V_2|$.

$$|\mathbb{E}V_2| \leq \beta\left(\alpha + \frac{\beta}{2}\right)\mathbb{E}\|z\|_2\,|\mathrm{e}_{i_*}^\top U^2 z|$$

$$\leq \beta\left(\alpha + \frac{\beta}{2}\right)\mathbb{E}\|z\|_2\,u_{i_*}^2|z_{i_*}|$$

$$\leq \beta\left(\alpha + \frac{\beta}{2}\right)\left(\mathbb{E}\|z\|_2^2 u_{i_*}^2\right)^{1/2}\Big(\underbrace{\mathbb{E}u_{i_*}^2 z_{i_*}^2}_{\triangleq V_3}\Big)^{1/2}. \quad \text{(Cauchy-Schwarz)} \quad (47)$$

For $V_3$, we can use an argument similar to Eq. (36) where we distribute the influence of the maximum coordinate over the $d$ coordinates.

$$V_3 = \mathbb{E}u_{i_*}^2 z_{i_*}^2 = \mathbb{E}\sum_{i=1}^d u_{i_*}^2 z_i^2 \mathbb{1}\{i_* = i\}$$

$$= \sum_{i=1}^d \mathbb{E}\left[u_{i_*}^2 C_{ii}^2 u_i^2 \mathbb{1}\{i_* = i\}\right]$$

$$= \sum_{i=1}^d C_{ii}^2 \mathbb{E}\left[u_{i_*}^4\right]\mathbb{E}[\mathbb{1}\{i_* = i\}] \quad (u_{i_*} \perp\!\!\!\perp i_*)$$

$$= \sum_{i=1}^d C_{ii}^2 \mathbb{E}\left[u_{i_*}^4\right]\mathbb{P}[i_* = i]$$

$$= \frac{1}{d}\mathbb{E}\left[u_{i_*}^4\right]\|C\|_\mathrm{F}^2$$

$$= \frac{1}{d}\left(\mathbb{E}u_{i_*}^4\right)\mathbb{E}\|z\|_2^2. \quad (48)$$

The last equality follows by applying Lemma A.2 to the identity $\mathbb{E}\|z\|_2^2 = \mathbb{E}u^\top C^\top C u$. By applying Eq. (48) into Eq. (47), we can now notice that $V_2$ decreases by a factor of $\mathbb{E}u_{i_*}^4/d$.

$$\mathbb{E}V_2 \geq -\beta\left(\alpha + \frac{\beta}{2}\right)\frac{\mathbb{E}\left[u_{i_*}^4\right]}{d}\sqrt{\mathbb{E}u_{i_*}^2\|z\|_2^2}\sqrt{\mathbb{E}\|z\|_2^2}$$

$$\geq -\beta\left(\alpha + \frac{\beta}{2}\right)\frac{\mathbb{E}\left[u_{i_*}^4\right]}{d}\mathbb{E}\left[u_{i_*}^2\|z\|_2^2\right]. \quad \text{(Assumption 2.2)}$$

It is clear that $V_2$ vanishes as $d \to \infty$.

Applying the lower bound on $V_2$ into Eqs. (45) and (46), we have

$$\mathbb{E}\left\|U\int_0^1 H_\mathrm{worst}(z^w)(z - \bar{z})\mathrm{d}w\right\|_2^2$$

$$\geq \frac{\beta^2}{4}\mathbb{E}\left[u_{i_*}^2\|z\|_2^2\right] - \beta\left(\alpha + \frac{\beta}{2}\right)\frac{\mathbb{E}\left[u_{i_*}^4\right]}{d}\mathbb{E}\left[u_{i_*}^2\|z\|_2^2\right]$$

$$= \left\{\frac{\beta^2}{4} - \left(\alpha\beta + \frac{\beta^2}{2}\right)\frac{\mathbb{E}\left[u_{i_*}^4\right]}{d}\right\}\mathbb{E}u_{i_*}^2\|z\|_2^2$$

$$= \left\{ \frac{(L-\mu)^2}{16} - \left( \frac{L^2 - \mu^2}{4} + \frac{(L-\mu)^2}{8} \right) \frac{\mathbb{E}\left[u_{i_*}^4\right]}{d} \right\} \mathbb{E} u_{i_*}^2 \|z\|_2^2 \qquad \text{(Eq. (44))}$$

$$\geq \left\{ \frac{(L-\mu)^2}{16} - \left( \frac{L^2 - \mu^2}{4} + \frac{L^2 + \mu^2}{4} \right) \frac{\mathbb{E}\left[u_{i_*}^4\right]}{d} \right\} \mathbb{E} u_{i_*}^2 \|z\|_2^2 \qquad \text{(Young's inequality)}$$

$$= \left\{ \frac{(L-\mu)^2}{16} - \frac{L^2}{2} \frac{\mathbb{E} \max_{i=1,\ldots,d} u_i^4}{d} \right\} \mathbb{E} u_{i_*}^2 \|Cu\|_2^2 . \tag{49}$$

It remains to solve the expectation.

Let us decompose the events where the $i$th coordinate attains the maximum ($i_* = i$) or not ($i_* \neq i$) as done in Lemma 4.1.

$$\mathbb{E} u_{i_*}^2 \|Cu\|_2^2 = \mathbb{E}\left[ \sum_{i=1}^d C_{ii} u_i^2 u_{i_*}^2 \right]$$

$$= \sum_{i=1}^d C_{ii}^2 \left\{ \mathbb{E}\left[ u_i^2 u_{i_*}^2 \mathbb{1}_{i_* = i} \right] + \mathbb{E}\left[ u_i^2 u_{i_*}^2 \mathbb{1}_{i_* \neq i} \right] \right\}$$

$$\geq \sum_{i=1}^d C_{ii}^2 \mathbb{E}\left[ u_i^2 u_{i_*}^2 \mathbb{1}_{i_* \neq i} \right] .$$

We are left with the expectation over the event $i_* \neq i$. For the upper bound in Lemma 4.1, the expectation was solved by noticing that $u_i^2$ and $u_{i_*}^2$ can be made independent after upper bounding the indicator. For a lower bound, however, breaking up the expectation for $u_i^2$ and $u_{i_*}^2$ is more involved.

$$\mathbb{E}\left[ u_i^2 u_{i_*}^2 \mathbb{1}_{i_* = i} \right] = \mathbb{E}\left[ u_i^2 \max_{j \neq i} u_j^2 \mathbb{1}_{i_* = i} \right]$$

$$= \mathbb{E}\left[ u_i^2 \max_{j \neq i} u_j^2 \mathbb{1}\left\{ u_i^2 < \max_{j \neq i} u_j^2 \right\} \right] . \tag{50}$$

By introducing a free variable $t > 0$, we can break up the indicator

$$\mathbb{1}\left\{ u_i^2 < \max_{j \neq i} u_j^2 \right\} \geq \mathbb{1}\left\{ u_i^2 < \max_{j \neq i} u_j^2, \, t < \max_{j \neq i} u_j^2 \right\}$$

$$\geq \mathbb{1}\left\{ u_i^2 < t, \, \max_{j \neq i} u_j^2 > t \right\}$$

$$= \mathbb{1}\left\{ u_i^2 < t, \right\} \mathbb{1}\left\{ \max_{j \neq i} u_j^2 > t \right\} . \tag{51}$$

This then allows the expectation to break up between terms depending on $u_i^2$ and $\max_{j \neq i} u_j$, which is the independence that we were after. That is, applying Eq. (51) to Eq. (50),

$$\mathbb{E}\left[ u_i^2 u_{i_*}^2 \mathbb{1}_{i_* = i} \right] \geq \mathbb{E}\left[ u_i^2 \max_{j \neq i} u_j^2 \mathbb{1}\left\{ u_i^2 < t, \right\} \mathbb{1}\left\{ \max_{j \neq i} u_j^2 > t \right\} \right]$$

$$= \mathbb{E}\left[ u_i^2 \mathbb{1}\left\{ u_i^2 < t \right\} \right] \mathbb{E}\left[ \max_{j=1,\ldots,d-1} u_j^2 \mathbb{1}\left\{ \max_{j=1,\ldots,d-1} u_j^2 > t \right\} \right]$$

$$= \mathbb{E}\left[ u_i^2 \mathbb{1}\left\{ u_i^2 < t \right\} \right] \left( \mathbb{E}\left[ \max_{j=1,\ldots,d-1} u_j^2 \right] - \mathbb{E}\left[ \max_{j=1,\ldots,d-1} u_j^2 \mathbb{1}\left\{ \max_{j=1,\ldots,d-1} u_j^2 \leq t \right\} \right] \right)$$

$$\geq \left( \int_0^t \mathbb{P}\left[ u_i^2 > s \right] \mathrm{d}s \right) \left( \mathbb{E}\left[ \max_{j=1,\ldots,d-1} u_j^2 \right] - t \right) .$$

Notice that the function $(t, \varphi) \mapsto \int_0^t \mathbb{P}\left[ u_i^2 > s \right] \mathrm{d}s$ is strictly positive as long as $t > 0$ and only dependent on $t$ and the base distribution $\varphi$.

We now obtain our final result by combining the results into Eq. (49). With explicit constants,

$$\mathbb{E}\left\|\int_0^1 H_{\text{worst}}(\mathbf{z}^w)(\mathbf{z}-\bar{z})\mathrm{d}w\right\|_{U^2}^2 \geq \left\{\frac{(L-\mu)^2}{4} - \frac{L^2}{2}\frac{\mathbb{E}\max_{i=1,\dots,d} u_i^4}{d}\right\}$$

$$\times \left(\int_0^t \mathbb{P}[u_i^2 > s]\mathrm{d}s\right)\left(\mathbb{E}\left[\max_{i=1,\dots,d-1} u_i^2\right] - t\right)\|C\|_{\text{F}}^2.$$

Substituting $c(t,\varphi) \triangleq \int_0^t \mathbb{P}[u_i^2 > s]\mathrm{d}s$ into this yields the stated result. $\qquad\square$

