# OpenReview forum: "Nearly Dimension-Independent Convergence of Mean-Field Black-Box Variational Inference"
_NeurIPS.cc/2025/Conference — NeurIPS 2025 poster_

### Official Review · Reviewer_Hyeq · 2025-06-25

**Clarity:** 3
**Significance:** 3
**Originality:** 3
**Rating:** 5
**Confidence:** 4

**Summary:**

This paper analyzes the rate of convergence of a particular optimization algorithm (stochastic proximal gradient descent using reparameterization gradients and a particular step size schedule) applied to mean-field variational inference for problems with strongly log-concave and log-smooth likelihoods using location-scale variational families. The main result is that the rate of convergence only depends weakly on the dimension of the problem: for variational families generated by sub-exponential random variables the dependence is $O(\log d)$, for variational families with $k$ finite moments it is $O(d^{2/k})$.  The authors then provide some justification that similar proof techniques could not result in a better bound.  The main ingredient in the proof is a careful analysis of the variance of the gradient in this setup, but the authors also use a number of additional clever tricks like a carefully chosen step size schedule.

**Questions:**

n/a

**Ethical Concerns:**

["NO or VERY MINOR ethics concerns only"]

**Final Justification:**

Based on my reading, the other reviews, and the authors' rebuttal to all the reviews, I feel that this is an exceptionally clear, well-written, and interesting paper.  As such, I feel that it is a clear "Accept".  There are practical limitations to implementing the results in the paper (e.g., the results only apply to very specific algorithmic choices), which makes the paper a bit too purely theoretical in my opinion to be useful to an extremely broad community, preventing me from choosing "Strong Accept".  Nevertheless, I enjoyed the paper and think it will make a strong addition to the conference.

**Limitations:**

yes

**Quality:**

4

**Strengths And Weaknesses:**

Strengths:
* Overall I thought that the paper was really clear. The exposition and proofs were generally easy to follow (although see below) and well-motivated.
* The main result is very interesting and suggests that mean-field BBVI should scale to higher dimensional problems more gracefully (at least computationally).
* The results could be useful in practice (e.g., setting step sizes or choosing a variational family to balance the ability to fit heavier tails with a worse scaling in dimension).

Weaknesses (all minor):
* There is a lot of emphasis on the scaling being nearly dimension-free, which -- unless I'm missing something --  obscures that some of the assumptions likely do not scale sub-linearly in the dimension.  For example, it would be reasonable to expect the Frobenius norm of the optimal scale matrix, $\Vert C_* \Vert_F^2$, to grow linearly in the dimension.  I don't think this is a major issue, but some discussion of how the other parameters in the bound are likely to grow with dimension in practice is warranted.
* It would be nice if the authors could discuss how relevant (at least in terms of the intuition) the authors think their results are to practical implementations.  For example, not all implementations use proximal SGD, many implementations use parameterizations other than the linear parameterization, the optimal step size and step size schedule depend on generally unknown parameters and so cannot be set like this in practice and so on.  Proofs in these other cases are obviously outside the scope of the paper, but it would be helpful if the authors could discuss which of their assumptions are for technical purposes and which they think are likely actually required of algorithms.
* I found the proof of Proposition 4.2 hard to follow.  In particular, it was unclear which parts of the proof are just for developing intuition for how they chose $H_{worst}$ and which parts of the proof are actually rigorous.  Relatedly, why is there a $z^w$ in the middle of the display after line 911? (And similarly in the second line of the display after line 921)?

Additional comments and typos:
* I think the step size schedule has a nice interpretation -- $t_*$ is chosen so that for the first $t_*$ steps the noise in the gradient is negligible relative to the ``signal''.  As a result, one can use a constant step size as if the gradients were noise free, this also allows one to geometrically converge from the initialization to $\lambda$ that are close enough to the optimum such that the noise in the gradients begins to dominate.  After this, one is in a regime that is more SGD-like and so a step-size like $1/t$ is required (c.f., Robbins-Monro).
* I found the phrasing around the formal Assumptions in the main text to be somewhat confusing on my first read.  For example "Assumptions" 2.1 and 2.2 read more like definitions, and the actual assumption is that various functions have those properties.  Relatedly, it was unclear throughout when "Assumptions" were being stated in the general case vs. being actual assumptions about the problem at hand.  For example, I think Assumption 2.7 should always hold for mean-field VI with location-scale families with linear parameterizations. I understand that it is being used to set up a more general result, but some changes to the wording might make that clearer.
* It's up to the authors, but Equation (16) appears in, I think, 3 different proofs, and I think it would be cleaner if that were made to be another Lemma in Appendix A instead of needing to refer back to an equation in the middle of a proof.
* Line 37: “parameterization gradient” —> “reparameterization gradient”
* Line 95: Shouldn't "diag" should be "vec"? $z$ is vector valued and $C$ is a matrix, so $Cu$ as defined would also be a $d\times d$ matrix, I think.
* Line 109: Latex typo $Q$ --> $\mathcal{Q}$
* Line 222: $q_T$ --> $q_{\lambda_T}$
* Line 253: missing reference in "of the result by , states that the highest"
* Line 290: second $\lambda$ should be $\lambda'$
* Line 326: "difficult to verify" --> "are difficult to verify"
* Line 362: "should easily be combined" --> "should be easy to combine"
* Line 669: second $\lambda$ should be $\lambda'$
* Line 735: "straightly" --> "straightforwardly"
* Not that it matters, but I believe the first line on p. 21 can be replaced with an equality because it's just algebra
* Statement of Lemma B.1: assumption should be on $\widehat{\nabla f}$ not $\nabla f$ (appears in both statements of Lemma B.1)
* "Equality" after line 704 should be $\le$
* After line 714 is presumably missing a $\le \epsilon$
* I believe the equations after lines 796 and 797 are missing terms that correspond to the function $g$ defined in the Theorem.
* Second line in the display after line 822: should be $X_i > t$ not $X_i > s$ in the domain of the indicator
* Norm on right hand side of first line of display after line 891 is missing a subscript 2.
* In both statements of Proposition 4.2 "location-family" --> "location-scale family"

---

> ### Author Rebuttal · Authors · 2025-07-30
>
> Thank you for your review!
>
> > There is a lot of emphasis on the scaling being nearly dimension-free, which -- unless I'm missing something -- obscures that some of the assumptions likely do not scale sub-linearly in the dimension. For example, it would be reasonable to expect the Frobenius norm of the optimal scale matrix, ${\lVert C_* \rVert}_{\mathrm{F}}$, to grow linearly in the dimension. I don't think this is a major issue, but some discussion of how the other parameters in the bound are likely to grow with dimension in practice is warranted.
>
> Thank you for the suggestion. Indeed, the term dimension-free appears to vary across communities and may not be clear to all readers. In the submission, we tried to avoid the ambiguity by stating “explicit dimension dependence” in both the abstract and the introduction, but not everywhere in the introduction. As such, in the next version, we will spell out “explicit” more often in the introduction and add a mention about implicit dimension dependence in both the introduction and results sections.
>
> > It would be nice if the authors could discuss how relevant (at least in terms of the intuition) the authors think their results are to practical implementations. For example, not all implementations use proximal SGD, many implementations use parameterizations other than the linear parameterization, the optimal step size and step size schedule depend on generally unknown parameters and so cannot be set like this in practice and so on. Proofs in these other cases are obviously outside the scope of the paper, but it would be helpful if the authors could discuss which of their assumptions are for technical purposes and which they think are likely actually required of algorithms.
>
> Indeed, the design space is large, and the algorithm design we chose in the paper does not exactly match practical implementations. In terms of parametrizations, we briefly mention Lines 124-128 that alternative “non-linear parametrizations” are commonly used. However, as mentioned, these usually rule out obtaining convergence guarantees [1] without setting a compact domain [2]. For optimization algorithms outside of proximal SGD, projected SGD is an obvious candidate. However, as mentioned in Lines 145-148, this requires setting an arbitrary compact domain, and both the analysis and convergence guarantees are analogous up to a factor of x2. (See [3].) However, we agree that a more holistic investigation and discussion over all these design choices would be useful for the community. For this, we plan on writing a paper performing an empirical comparison that would hopefully act as a guide to the various pros and cons of these design elements.
>
> For setting the step size schedule and initial step size, we agree that a discussion about practical considerations would be useful. We will mention the fact that the act of parameter tuning can be seen as identifying the step size (schedules) that depend on the unknown problem constants.
>
> > I found the proof of Proposition 4.2 hard to follow. In particular, it was unclear which parts of the proof are just for developing intuition for how they chose $H_{\mathrm{worst}}$ and which parts of the proof are actually rigorous. Relatedly, why is there a $z^{w}$ in the middle of the display after line 911? (And similarly in the second line of the display after line 921)?
>
> Thank you for pointing this out! We agree that the use of $z^w$ here is extremely confusing. The original intent was to convey that the result of the integral $\int^1_0 H_{\mathrm{worst}}(z^w) (z - \bar{z}) \mathrm{d}w$ is the same as the pointwise evaluation of $H_{\mathrm{worst}}(z^w) (z - \bar{z})$ with respect to any $w \in [0, 1]$, but the notation is inappropriate. We will fix this in the next version. In terms of the organization of the proof, we will add more signpost text to clearly demarcate where the requirements for the worst-case example are stated and the actual proof begins.
>
> > I found the phrasing around the formal Assumptions in the main text to be somewhat confusing on my first read. For example "Assumptions" 2.1 and 2.2 read more like definitions, and the actual assumption is that various functions have those properties.
>
> Thank you for the suggestion. We agree that “Definition” would have been better here, and have in fact changed this shortly after our initial submission.
>
> > For example, I think Assumption 2.7 should always hold for mean-field VI with location-scale families with linear parameterizations. I understand that it is being used to set up a more general result, but some changes to the wording might make that clearer.
>
> Thank you for the suggestion. We agree that some text to explain the relevance to VI would have been better here. We will improve this in the next version.
>
> And thank you for spotting the various typos in the proofs! We will make sure to fix these in the next version.
>
> 1. Kim, K., Oh, J., Wu, K., Ma, Y., & Gardner, J. (2023). On the convergence of black-box variational inference. Advances in Neural Information Processing Systems, 36, 44615-44657.
> 2. Hotti, A. M., Van der Goten, L. A., & Lagergren, J. (2024, April). Benefits of non-linear scale parameterizations in black box variational inference through smoothness results and gradient variance bounds. In International Conference on Artificial Intelligence and Statistics (pp. 3538-3546). PMLR.
> 3. Domke, J., Gower, R., & Garrigos, G. (2023). Provable convergence guarantees for black-box variational inference. Advances in neural information processing systems, 36, 66289-66327.

---

> > ### Comment · Reviewer_Hyeq · 2025-08-04
> >
> > Thank you for the detailed response.  All of my initial comments were minor (I hope they were useful), and my initial assessment that this paper is a clear accept remains unchanged.

---

### Official Review · Reviewer_FijK · 2025-06-29

**Clarity:** 3
**Significance:** 3
**Originality:** 3
**Rating:** 5
**Confidence:** 3

**Summary:**

In this paper, the authors developed a new theoretical convergence guarantee for black-box variational inference when using mean-field location-scale families. They show that by using stochastic proximal gradient descent, it can achieve a convergence rate that is nearly independent of dimension for target distributions that are log-smooth and strongly log-concave. Moreover, authors have analyzed the variance of the reparameterization gradient and showed that the bound on the gradient variance cannot be improved using only spectral bounds on the Hessian of the target log-density.

**Questions:**

1. Typo in lines 144 & 185: It looks like the acronym "PSGD" is a typo. Should it be SPGD instead?
2. Should the matrix $C$ in Definition 2.6 be required to be positive definite and diagonal?

**Ethical Concerns:**

["NO or VERY MINOR ethics concerns only"]

**Final Justification:**

This is a technically sound and well-argued paper that addresses an important topic. The authors’ rebuttal addresses my earlier concerns. While the lack of illustrative experiments limits the immediate empirical validation, I understand the challenges involved and find the theoretical contributions compelling. Overall, I remain confident in my original assessment, "accept", and believe the paper makes a valuable contribution.

**Limitations:**

All limitations are addressed.

**Paper Formatting Concerns:**

I did not see any major formatting issues in the paper.

**Quality:**

4

**Strengths And Weaknesses:**

Strengths:
This paper is well-written and clearly structured. The inclusion of proof sketches is helpful for readers. The paper demonstrates that the iteration complexity of mean-field location-scale families, which are commonly used in practice, can be improved to nearly a dimension-independent rate, representing an advancement over previous work. I believe this is a significant contribution to the variational inference community.

Weaknesses:
The paper is highly theoretical and lacks numerical experiments or synthetic illustrations. I think including illustrations would help the reader follow the concepts more easily. While the paper focuses on the reparameterization gradient and provides a technical justification for its use, it does not adequately explain why this estimator is prioritized over alternatives like the score function gradient. Providing additional context or justification could strengthen the relevance of the paper's contributions.

---

> ### Author Rebuttal · Authors · 2025-07-30
>
> Thank you for your review!
>
> > The paper is highly theoretical and lacks numerical experiments or synthetic illustrations. I think including illustrations would help the reader follow the concepts more easily.
>
> Thank you for the suggestion. We are open to suggestions for any illustrations or experimental results that would help readers understand the technical content of the paper. Please feel free to suggest anything the reviewer would find useful!
>
> > While the paper focuses on the reparameterization gradient and provides a technical justification for its use, it does not adequately explain why this estimator is prioritized over alternatives like the score function gradient.
>
> As mentioned in Lines 154-155, we focus on the reparametrization gradient as it is empirically known to perform much better than the score gradient [1,2]. This fact has also been theoretically established in [3] for Gaussian targets. In addition, as mentioned in Lines 76-78, most probabilistic programming frameworks use the reparametrization gradient by default whenever applicable. Therefore, we believe it is fair to say that the reparametrization gradient is the most popular choice in practice. However, we do agree that the score gradient also warrants some attention in terms of a general theoretical analysis. But this would be worth its own paper.
>
> > Providing additional context or justification could strengthen the relevance of the paper's contributions.
>
> Thank you for the suggestion. As mentioned in Section 1, a quantitative convergence analysis of mean-field BBVI is practically relevant regarding the statistical-computational trade-off. That is, it is useful to know how long it would take to obtain a full-rank approximation compared to a mean-field approximation. Our paper implies that, in the high-dimensional regime, one should roughly expect a factor of $d$ difference. (See also the reply to Reviewer P294.)
>
> > Typo in lines 144 & 185: It looks like the acronym "PSGD" is a typo. Should it be SPGD instead?
>
> Yes, thank you for spotting this! It should have been SPGD.
>
> > Should the matrix $C$ in Definition 2.6 be required to be positive definite and diagonal?
>
> This is somewhat a matter of convention. Mathematically speaking, $C$ only needs to be invertible to result in a well-defined distribution since the resulting covariance $CC^{\top}$ will be positive definite even if $C$ has negative eigenvalues. However, it is indeed common to define $C$ to have strictly positive eigenvalues, since, in the 1-dimensional, allows to interpret $C$ as a standard deviation. For our purpose, this requirement isn't necessary for defining location-scale families, so we didn't mention it Definition 2.6. Though we should clarify that $C$ at least needs to be invertible. We will fix this in the next version and thank you for pointing this out.
>
> 1. Titsias, M., & Lázaro-Gredilla, M. (2014, June). Doubly stochastic variational Bayes for non-conjugate inference. In International conference on machine learning (pp. 1971-1979). PMLR.
> 2. Kucukelbir, A., Tran, D., Ranganath, R., Gelman, A., & Blei, D. M. (2017). Automatic differentiation variational inference. Journal of machine learning research, 18(14), 1-45.
> 3. Xu, M., Quiroz, M., Kohn, R., & Sisson, S. A. (2019, April). Variance reduction properties of the reparameterization trick. In The 22nd international conference on artificial intelligence and statistics (pp. 2711-2720). PMLR.

---

> > ### Comment · Reviewer_FijK · 2025-08-02
> > **Thank you for the response**
> >
> > Thank you for providing a detailed rebuttal. Regarding the lack of illustrations, I understand the challenges in designing meaningful experiments, especially considering your explanation to the reviewer P294. I agree with the authors' justification for focusing on the reparameterization gradient, and I appreciate the thorough explanation regarding matrix $C$. Overall, I remain optimistic about the contribution and would like to keep my current score.

---

### Official Review · Reviewer_P294 · 2025-07-01

**Clarity:** 4
**Significance:** 3
**Originality:** 2
**Rating:** 5
**Confidence:** 2

**Summary:**

In this work the authors investigate the problem of optimizing black box variational inference, in particular regularized problems which use location (a $d$-dimensional vector $m$ designating the "center" of the distribution) and scale ( a $d\times d$-dimensional matrix $C$, which linearly transforms the distribution, may be enforced to be diagonal, low rank plus identity, etc) parameters, which are contained in $\lambda$. In such problems,  it is theoretically known to that estimate to $\epsilon$ accuracy is known to take order $d\kappa^2 \epsilon^-1$. It has been conjectured that this dependence on $d$ is no so bad. Indeed for the *mean field* setting where the matrix is enforced to be diagonal, this improves to order $\sqrt{d}\kappa^2\epsilon^-1$, yet it is thought that this dependence on $d$ can be further improved.

This paper presents results showing that estimating $\lambda$ using a proximal stochastic gradient descent algorithm with correct step sizes can achieve much better dependences on $d$, with $\log(d)$ dependence in the case of a sub-Gaussian family, and $d^{2/k}$ when $k$ moments exist for the distribution (so $d^{2/(2-\nu)}$ for Student-t with $\nu$ degrees of freedom).

**Questions:**

I don't have any particularly deep questions regarding the paper.

* I wonder if it would be possible to verify the results experimentally?
* Are there any concrete useable/practical lessons from this paper, other than one need not worry so much about dimensionality when the data is expected to be not heavy tailed.
* Again the main point keeping me from a strong accept is my lack of certainty regarding whether the impact, originality or usefulness is good enough to warrant the top reviewing score. This may just be because I am not super comfortable with the topic. Anything you could do here to elaborate for a non-expert would be helpful.

**Ethical Concerns:**

["NO or VERY MINOR ethics concerns only"]

**Final Justification:**

I think accept is warrated for the reasons given in my review. I don't think this is quite strong enough for a spotlight paper for the reasons mentioned. A score of 5 is appropriate.

**Limitations:**

I think the limitations are pretty clear from the exposition in the paper that indicate where their results apply.

**Paper Formatting Concerns:**

None.

**Quality:**

4

**Strengths And Weaknesses:**

This paper shines light on a very simple yet fundamental technique in a statistical method that is quite popular recently. Understanding how these estimators work in high dimensions is of interest. I found that paper pleasant to read, even though I feel as though I couldn't fully grasp every part, which is probably unavoidable due to the topicality.

I've scanned the proofs and they also look very nice. They seem well organized with clear justifications provided at every step and signposts for the reader to help guide the proof logic.

For weaknesses, I don't have any significant issues with the work. The paper is purely theoretical result on a phenomenon that was already suspected, on a topic that isn't as mainstream, as some at NeurIPS, with no clear practical impact. Hence I am a bit reluctant to give a strong accept. This line of work decently outside of my field of expertise so its possible that I'm missing something here.

Small errors:

* line 144: PSGD is never defined, and the sentence "Instead of using PSGD, one can also use projected SGD" is a bit confusing since its easy to think that PSGD refers to projected SGD. I think this may actually be an issue in other parts of the paper. Please just make sure the nomenclature is clear and consistent.

* Prop 2.11 (and elsewhere): I don't really like the $T \ge O(...)$. Maybe $\Omega$ is better?

* line 193 and 196: "anytime" vs "any-time"

* line 253: missing citation

---

> ### Author Rebuttal · Authors · 2025-07-30
>
> Thank you for your review!
>
> > line 144: PSGD is never defined, and the sentence "Instead of using PSGD, one can also use projected SGD" is a bit confusing since its easy to think that PSGD refers to projected SGD.
>
> We apologize for the confusion, this is a typo and should have been SPGD for stochastic proximal gradient descent.
>
> > Prop 2.11 (and elsewhere): I don't really like the $T \geq O(\ldots)$. Maybe $T \geq \Omega(\ldots)$ is better?
>
> Thank you for the suggestion. Indeed, this is a common point of confusion, but in this case, big-O is correct, whereas big-$\Omega$ is not quite. This is because $T \geq O(\ldots) \Rightarrow {\lVert \lambda_0 - \lambda_* \rVert}_2^2 \leq \epsilon$ implies that “the smallest $T$ one can choose to guarantee the $\epsilon$-accuracy condition grows no faster than $\ldots$.” Therefore, this is an “upper bound” on the sufficient number of iterations one needs to take. On the other hand, if one uses big-$\Omega$, it would read, “the smallest $T$ one can choose grows faster than $\ldots$,” which is a statement of necessity rather than sufficiency.
>
> > I wonder if it would be possible to verify the results experimentally?
>
> Thank you for the suggestion. We agree that empirical verification of theoretical results is worth doing. Now, the fact that the mean-field approximation results in better scaling has been known empirically for a long time. Therefore, we found that simply showing that the mean-field approximation scales well is not empirically meaningful. Instead, it would be most interesting to verify that the worst-case $O(\log d)$ dimensional scaling can be observed in practice. However, as noted in Remark 4.3, the theoretical worst-case example that achieves this $O(\log d)$ scaling may not exist. We tried various informed guesses for $f$ to see if we observe an increasing scaling with respect to $d$. However, all examples either resulted in a constant scaling or even decreasing (!) scaling with respect to $d$. Therefore, we couldn’t design an interesting enough experiment to verify the results in the paper.
>
> > Are there any concrete useable/practical lessons from this paper, other than one need not worry so much about dimensionality when the data is expected to be not heavy tailed.
>
> > Again the main point keeping me from a strong accept is my lack of certainty regarding whether the impact, originality or usefulness is good enough to warrant the top reviewing score. This may just be because I am not super comfortable with the topic. Anything you could do here to elaborate for a non-expert would be helpful.
>
> This is an important point. Our results say a bit more than one needs to worry about dimensionality. As mentioned in Section 1, we believe VI is the most relevant for obtaining a controllable statistical-computational trade-off. That is, we would like to know how much speed up we can obtain by giving away some amount of statistical accuracy. For this, a quantitative convergence analysis is necessary for variational families of various degrees of expressivity. With this in mind, our quantitative convergence analysis implies that, in the high-dimensional regime, our results suggest moving from a full-rank family to a mean-field family will require running SGD roughly $d$ times longer with a smaller step size. For variational families with an expressivity in between mean-field and full-rank, we can expect to see a dimensional scaling in between $O(\log d)$ and $O(d)$, where we can use the $O(\log d)$ result as a baseline for estimating the amount of necessary computation time. (See also the last paragraph of Section 5.2.)

---

> > ### Comment · Reviewer_P294 · 2025-08-02
> > **Maintain Score**
> >
> > I appreciate the author’s response. As I said before, I think this is a good paper and a clear *accept*. I don’t think it quite rises to the level of a spotlight—perhaps that could have been possible with more compelling experimental evidence, though I understand that may not be feasible. Nonetheless, a solid paper!

---

### Official Review · Reviewer_S4iz · 2025-07-03

**Clarity:** 4
**Significance:** 3
**Originality:** 3
**Rating:** 5
**Confidence:** 4

**Summary:**

This paper proves that mean field VI converges at an $O(log d)$, where $d$ is the dimension of the parameters, improving on an $O(d)$ rate for full-rank VI. Previously, it was known that the rate could at most be on the order of the square root of d, and faster convergence was speculated but not proven. If the Hessian of the target density is constant, there is no dimension dependence. To obtain this result, there are assumptions about the regularity of the target density (log-concave and log-smooth), the variational family (mean field location-scale family with sub-Gaussian tails), optimization (proximal SGD), and the gradient estimator (reparameterization trick). The assumption of sub-Gaussian tails can be relaxed, leading to, for example in the student-t case, a $O(d^2/(\nu-2))$ dependence, where $\nu$ is the degrees of freedom.

**Questions:**

- Theorem 3.2 states that there exists a switching time and initial step size for which the result holds. Can you comment on the practical significance of having to choose these values?

Small:
- Missing citation on line 253
- Should be $\lambda’ = (m’, C’)$ in line 290
- In the equation after line 290, I believe it should be an inequality right after $V_{scale}$

**Ethical Concerns:**

["NO or VERY MINOR ethics concerns only"]

**Final Justification:**

This is a clearly-written, rigorous theoretical paper, that I think is a clear accept. As some of the reviewers pointed out, the demonstration of the practical impact via experiments would add to the paper, but I also understand the authors response about the challenges in doing so and maintain my original assessment that experiments are not needed for the paper to be accepted.

**Limitations:**

Yes

**Paper Formatting Concerns:**

No concerns

**Quality:**

4

**Strengths And Weaknesses:**

This is an exceptionally well-written paper, with a detailed overview of related work and rigorous proofs of all claims. The assumptions are standard and discussed carefully. The contribution is important because it may justify the use of mean field VI for high-dimensional problems.

Section 4, and the proof sketch of lemma 4.1, provides a nice explanation of the challenges in the proof and why past works obtained looser bounds.

I think the lack of experiments is fine for this kind of paper, but it would be nice to understand the practical significance for common problems.

As the authors remark, it had been speculated that a faster convergence happens in the mean field case, so one weakness is that the result doesn’t seem surprising, but there is value in identifying and proving the rate as this paper does. The proof strategy also seems like a nice contribution.

---

> ### Author Rebuttal · Authors · 2025-07-30
>
> Thank you for your review!
>
> > Theorem 3.2 states that there exists a switching time and initial step size for which the result holds. Can you comment on the practical significance of having to choose these values?
>
> Yes. In practice, it is no secret that any application of SGD requires extensive tuning of the step size schedule. Indeed, Proposition 2.11 reflects this fact through the use of the two-step step size schedule in Eq. (4). Thus, in that sense, our theory is qualitatively in agreement with practice.
>
> Furthermore, in previous works [1], the switching time was selected only to depend on the condition number $\kappa$. This suggests that it is sufficient to adapt the step size schedule (or more precisely, the switching time) according to the target problem only. Proposition 2.11 demonstrates that, in order to further optimize the dependence on the initialization ${\lVert\lambda_0 - \lambda_*\rVert}_2$, it is necessary to also adapt the switching time according to the properties of the gradient estimator ($\sigma^2$). That is, our results suggest that whenever the gradient estimator is swapped, the step size schedule should also be adjusted accordingly to obtain faster convergence. We will make sure to mention this fact in the next version.
>
> 1. Gower, R. M., Loizou, N., Qian, X., Sailanbayev, A., Shulgin, E., & Richtárik, P. (2019, May). SGD: General analysis and improved rates. In International conference on machine learning (pp. 5200-5209). PMLR.

---

> ### Comment · Reviewer_S4iz · 2025-08-04
>
> Thank you for the reply to my question. I maintain that this paper is a clear accept.

---

### Decision · Program_Chairs · 2025-09-17

**Decision:**

Accept (poster)

**Comment:**

We thank the authors for their submission.

This work sheds theoretical light on the convergence of a commonly used variational approximation — location scale mean field families. Notably, the authors show that under certain conditions, convergence only weakly depends on the dimension of the parameters.  Reviewers agree that this is a well-written paper with a theoretically sound analysis of a popular algorithm, justifying its use and making more precise under what conditions/likelihoods may enjoy these theoretical guarantees.